# Discovering genetic interactions bridging pathways in genome-wide association studies

Gang Fang [1,6], Wen Wang [2,6], Vanja Paunic[2], Hamed Heydari [3], Michael Costanzo [3], Xiaoye Liu[2], Xiaotong Liu [2], Benjamin VanderSluis [2], Benjamin Oately[2], Michael Steinbach [2], Brian Van Ness[4], Eric E. Schadt [1], Nathan D. Pankratz [5], Charles Boone[3], Vipin Kumar[2] & Chad L. Myers [2]

Genetic interactions have been reported to underlie phenotypes in a variety of systems, but the extent to which they contribute to complex disease in humans remains unclear. In principle, genome-wide association studies (GWAS) provide a platform for detecting genetic interactions, but existing methods for identifying them from GWAS data tend to focus on testing individual locus pairs, which undermines statistical power. Importantly, a global genetic network mapped for a model eukaryotic organism revealed that genetic interactions often connect genes between compensatory functional modules in a highly coherent manner. Taking advantage of this expected structure, we developed a computational approach called BridGE that identifies pathways connected by genetic interactions from GWAS data. Applying BridGE broadly, we discover significant interactions in Parkinson's disease, schizophrenia, hypertension, prostate cancer, breast cancer, and type 2 diabetes. Our novel approach provides a general framework for mapping complex genetic networks underlying human disease from genome-wide genotype data.

[1] Department of Genetics and Genomic Sciences, Icahn School of Medicine at Mount Sinai, New York, NY 10029, USA. [2] Department of Computer Science and Engineering, University of Minnesota, Minneapolis, MN 55455, USA. [3] Donnelly Centre, University of Toronto, Toronto, ON M5S 3E1, Canada. [4] Department of Genetics, Cell Biology, and Development, University of Minnesota, Minneapolis, MN 55455, USA. [5] Department of Laboratory Medicine and Pathology, University of Minnesota, Minneapolis, MN 55455, USA. [6]These authors contributed equally: Gang Fang, Wen Wang. Correspondence and requests for materials should be addressed to G.F. (email: gang.fang@mssm.edu) or to V.K. (email: kumar@cs.umn.edu) or to C.L.M. (email: chadm@umn.edu)

Genome-wide association studies (GWAS) have been increasingly successful at identifying single-nucleotide polymorphisms (SNPs) with statistically significant association to a variety of diseases[1,2] and gene sets significantly enriched for SNPs with moderate association[3]. However, for most diseases, there remains a substantial disparity between the disease risk explained by the discovered loci and the estimated total heritable disease risk based on familial aggregation[4,5]. While there are a number of possible explanations for this "missing heritability", including many loci with small effects or rare variants[4], genetic interactions between loci are one potential culprit[5,6]. Genetic interactions generally refer to a combination of two or more genes whose contribution to a phenotype cannot be completely explained by their independent effects[5,7]. One example of an extreme genetic interaction is synthetic lethality where two mutations, neither of which is lethal on its own, combine to generate a lethal double mutant phenotype. Thus, genetic interactions may explain how relatively benign variation can combine to generate more extreme phenotypes, including complex human diseases[4,5,8]. Several studies have reported genetic interactions between specific variants in various disease contexts[7,9], and scalable computational tools have been developed for searching for interactions amongst SNPs[7,10]. However, systematic discovery of statistically significant genetic interactions on a genome-scale remains a major challenge. For example, a theoretical analysis estimated that ~500,000 subjects would be needed to detect significant genetic interactions under reasonable assumptions[5], which remains beyond the cohort sizes available for a typical GWAS study or even the large majority of meta-GWAS studies.

Genome-wide, reverse genetic screens in model organisms have produced rich insights into the prevalence and organization of genetic interactions[11,12]. Specifically, the mapping and analysis of the yeast genetic network revealed that genetic interactions are numerous and tend to cluster into highly organized network structures, connecting genes in two different but compensatory functional modules (e.g., pathways or protein complexes) as opposed to appearing as isolated instances[11,13]. For example, nonessential genes belonging to the same pathway often exhibit negative genetic interactions with the genes of a second nonessential pathway that impinges on the same essential function (Fig. 1a). Owing to their functional redundancy, the two different pathways can compensate for the loss of the other, and thus, only simultaneous perturbation of both pathways (e.g., A* and Y*) (Fig. 1a) results in an extreme loss of function phenotype, which could be associated with either increased or decreased disease risk. Importantly, the same phenotypic outcome could be achieved by several different combinations of genetic perturbations in both pathways (e.g., A-X, A-Z, B-X, B-Y, B-Z) (Fig. 1b). This model for the local topology of genetic networks, called the "between-pathway model" (BPM), has been widely observed in yeast genetic interaction networks[11,14]. Indeed, as many as ~70% of negative genetic interactions observed in yeast occur in BPM structures, indicating that genetic interactions are highly organized and this type of local clustering is the rule rather than the exception[13]. In addition to BPMs, combinations of mutations in genes within the same pathway or protein complex also tend to exhibit a high frequency of genetic interaction (Fig. 1b), a network structure referred to as a "within-pathway model" (WPM)[11,14]. Indeed, ~80% of essential protein complexes in yeast exhibit a significantly elevated frequency of within-pathway interactions[15]. In the context of human disease, a WPM may reflect an individual that inherits two variants in the same pathway, resulting in reduced flux or function of a particular pathway and an increase or decrease in disease risk.

The prevalence of BPM and WPM structures observed in the yeast global genetic network has important practical implications that can be exploited to explore disease-associated genetic interactions in humans based on GWAS data. Although tests to identify genetic interactions between specific SNP or gene pairs are statistically under-powered, we may be able to detect genetic interactions by leveraging the fact that pairwise interactions between genome variants are likely to cluster into larger BPM and WPM network structures similar to those observed in the yeast global genetic network. Indeed, other studies exploited similar structural properties to derive genetic interaction networks from phenotypic variation in a yeast recombinant inbred population[16].

Here, we present a new method, called BridGE (Bridging Gene sets with Epistasis), that leverages the expected between- and within-pathway structure of genetic interactions to discover them based on human population genetic data. We present results from application of this method to seven different diseases, with significant interactions discovered in six of the seven, along with extensive simulation results establishing the utility of the approach. We note that the method proposed here is broadly similar to previous approaches that have used gene-set enrichment or GO enrichment analysis to interpret SNP sets arising from univariate or interaction analyses[3,17] or aggregation tests for rare variants[18,19] (see Methods). Other existing approaches have also successfully identified interactions by reducing the test space for SNP–SNP pairs, either through knowledge or data-driven prioritization[20,21] (see Methods). However, to our knowledge, no existing method has been developed to systematically identify between-pathway genetic interaction structures based on human genetic data, which is the focus of this study.

## Results

**A new method for discovery of genetic interactions from GWAS.** We developed a method called BridGE (Bridging Gene sets with Epistasis) to explicitly search for coherent sets of SNP–SNP interactions within GWAS cohorts that connect groups of genes corresponding to characterized pathways or functional modules. Specifically, although many pairs of loci do not have statistically significant interactions when considered individually, they can be collectively significant if an enrichment of SNP–SNP interactions is observed between two functionally related sets of genes or within a functional coherent gene set (Fig. 1b). Thus, we imposed prior knowledge of pathway membership and exploited structural and topological properties of genetic networks to gain statistical power to detect genetic interactions that occur between or within defined pathways in GWAS associated with diverse diseases. Our algorithm identifies BPM structures, where two distinct pathways are "bridged" by several SNP-level interactions that connect them, as well as WPM structures, where interactions densely connect SNPs linked to genes in the same pathway, and finally, pathways whose SNPs show increased interaction density relative to the rest of SNPs in the network, though not coherently as in a BPM (referred to as "PATH" structures).

Our approach involves five main components, each of which is described briefly below (Fig. 1c and Supplementary Fig. 1; see Methods): (I) Data processing; (II) construction of a SNP–SNP interaction network; (III) binarization of the SNP–SNP network; (IV) scoring of BPMs and WPMs for enrichment of SNP–SNP interactions; (V) estimation of false discovery rate based on a hybrid permutation strategy. (I) For data processing, in addition to standard GWAS data processing procedures, BridGE controls for population substructure between the cases and controls to eliminate false discoveries due to population stratification[22]. Also, in dataset pre-processing, the input set of SNPs is pruned down to a subset of SNPs that are less likely in linkage disequilibrium (LD) with each other to avoid discovery of spurious BPM/WPM structures due to LD. (II) Construction of a SNP–SNP interaction

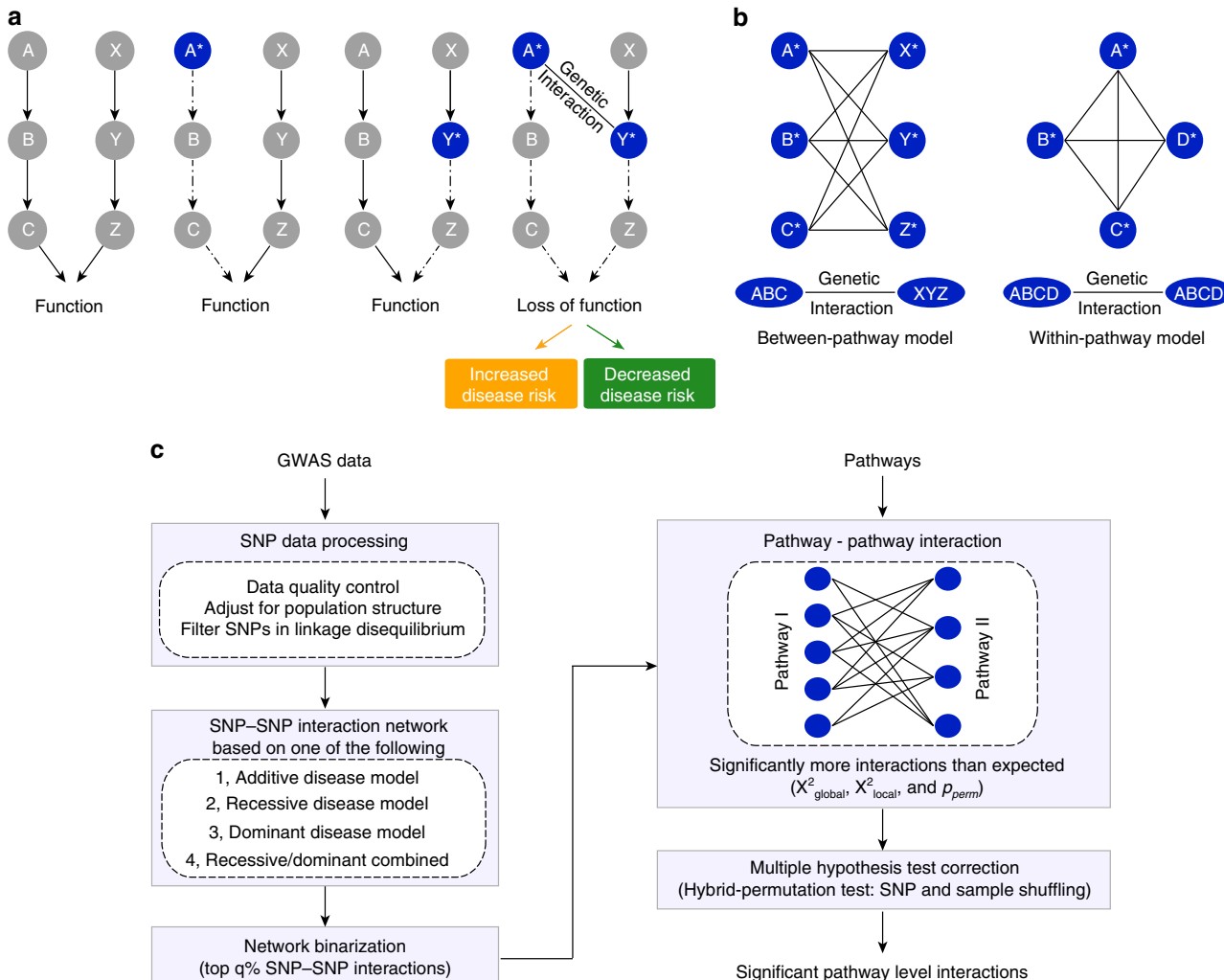

**Fig. 1** Between-pathway and within-pathway model of genetic interactions. **a** Two distinct pathways, A → B → C and X → Y → Z converge to regulate the same essential function. Independent genetic perturbations in either pathway (indicated by blue color with an asterisk) have little or no contribution to a phenotype, but combined perturbations in both pathways in the same individual result in a genetic interaction, leading to a loss of function phenotype that can be associated with either an increase or decrease in disease risk. **b** The bipartite structure of genetic interactions (between-pathway model) resulting from functional compensation between the two pathways shown in **a** and the quasi-clique structure of genetic interactions based on the within-pathway model. Genetic perturbations in any pair of genes across the two pathways or within the same pathway combine to increase or decrease disease risk. Edges indicate observed interactions at the gene–gene or SNP–SNP level. **c** Conceptual overview of the BridGE method for detecting genetic interactions from GWAS data

network: BridGE can use any one of four different disease models to compute a SNP–SNP interaction network: additive, recessive, dominant, or combined recessive and dominant. When run in the additive, recessive and dominant mode, BridGE tests all SNP pairs across the two pathways using the same disease model. In the combined recessive and dominant mode, BridGE integrates edges derived from multiple disease models, which allows for SNPs to interact in multiple ways as long as they still connect across the two pathways of interest. Each model encodes different assumptions and will result in a different, overall complementary, set of significant BPM/WPM structures at the end of the pipeline. The additive disease model was implemented as previously described[8], with SNP–SNP interaction scores derived from likelihood ratio tests comparing models with and without an interaction term[7]. Interactions based on recessive and dominant disease models are estimated using a hypergeometric-based metric that directly tests for disease association for individuals that are either homozygous (recessive and dominant models) or heterozygous (dominant only) for the minor allele at two loci of

interest, and compares the observed degree of association to the marginal effects of both loci. BridGE can be run in any of these disease model modes and also implements a combined dominant/recessive mode where interactions among SNP pairs are taken from the maximum (most significant) of the dominant and recessive models. (III) Binarization of the SNP–SNP network: The SNP–SNP network derived from a chosen disease model is thresholded by applying a lenient significance cutoff to all pairwise SNP–SNP interactions to generate a low-confidence, high-coverage network that is expected to contain a large number of false positive interactions, but enables the assessment of significance of SNP–SNP interactions collectively at the pathway level. The binarization threshold is determined by a pilot run that evaluates the SNP–SNP interaction network at a range of cutoffs with a limited number of permutations to establish preliminary significance estimates (see Methods). (IV) Scoring of BPMs and WPMs for enrichment of SNP–SNP interactions: after SNP–SNP network binarization, pathways are defined by curated functional standards[23–25], and these pathways (in the case of WPMs) and

pairs of pathways (for BPMs) are tested for enrichment of SNP–SNP pair interactions. The enrichment significance is measured by a permutation test ($p_{perm}$) derived from randomly shuffling of the SNP-pathway assignment (see Methods). In addition to discovering BPM and WPM structures, our approach also identifies individual pathways that have significantly elevated marginal density of SNP–SNP interactions even where the interaction partners do not necessarily have clear coherence in terms of pathways (PATH structures, see Methods). In this case, we are not identifying pathway–pathway interactions but simply assessing whether a particular pathway is a highly connected hub and associated with numerous SNP-level interactions. (V) Estimation of false discovery rate based on a hybrid permutation strategy: finally, to correct for multiple hypothesis testing and assess the significance of the candidate BPM, WPM, or PATH interactions, BridGE uses sample permutation to establish an FDR. Specifically, FDR is estimated by measuring the average number of discovered BPMs/WPMs from the random sample labels that achieve a higher significance level than the BPMs/ WPMs reported from the real sample labels relative to the total number of discoveries (i.e., FDR ~ average number of discoveries at greater significance in the random permutations/total number of discoveries in the real dataset). As the number of hypothesis tests performed for all possible pathways and all possible between-pathway combinations is substantially less than the number of tests for all possible SNP pairs ($\sim10^5$ as compared to $\sim10^{11}$), our power for discovering interactions relative to approaches that operate on individual SNP–SNP interactions is greatly increased. These five steps enabled us to extract statistically significant pathway-level interactions that can be associated with either increased risk of disease when pairs of minor alleles linked to two pathways occur more frequently in the diseased population or, conversely, decreased risk of disease when pairs of minor alleles annotated to two pathways occur more frequently in the control population. The code implementing our approach is available at http://csbio.cs.umn.edu/bridge.

**Discovery of interactions in a Parkinson's disease cohort.** We first applied BridGE to identify BPM genetic interactions in a genome-wide association study of Parkinson's disease (PD)[26], denoted as PD-NIA (Supplementary Data 1). Recent work estimated a substantial heritable contribution to PD risk across a variety of GWAS designs (20–40%)[27,28], and although a relatively large number of variants have been individually associated with PD, the loci discovered to date explain only a small fraction (6–7%) of the total heritable risk[27]. The PD-NIA cohort used in this analysis consists of 519 patients and 519 ancestry-matched controls after balancing the population substructure (see Methods). We compiled a collection of 833 curated gene sets (MSigDB Canonical pathways)[29] representing established pathways or functional modules from KEGG[23], BioCarta[24], and Reactome[25] (Supplementary Data 2) and found that 658 of these pathways were represented in the PD-NIA cohort after filtering based on gene-set size (minimum: 10 genes or SNPs, maximum: 300 genes or SNPs). After using SNP-pathway membership permutations (NP = 150,000) and sample permutations (NP = 10) to establish global significance and correct for the multiple hypotheses tested (see Methods), BridGE reported 23 significant BPMs involving 32 unique pathways at a false discovery rate (FDR) of ≤0.25 ($p_{perm} < 4.7 \times 10^{-5}$) using a combined disease model (Figs. 2a, 3, Supplementary Data 3, 4). For example, one of the highest confidence BPMs (FDR ≤ 0.05) was identified between the Golgi-associated vesicle biogenesis and FC epsilon receptor I (FcεRI) signaling pathways. More specifically, we observed 2281 SNP–SNP interactions between the vesicle biogenesis and FcεRI signaling gene

sets (Fig. 2b), a 1.5-fold enrichment relative to the expected number of SNP–SNP interactions (1510) based on the global density of the SNP–SNP interaction network (5%) and 1.3- and 1.2-fold enrichment given the marginal density of the two pathways (5.9% and 6.5%), respectively ($p_{perm} < 6.7 \times 10^{-6}$, Fig. 2c). Importantly, none of the individual SNPs-associated genes annotated to either the Golgi-associated vesicle biogenesis or FcεRI signaling were significant on their own after multiple hypothesis correction based on single-locus tests on this cohort (Fig. 2b). Furthermore, none of the individual SNP–SNP interactions between the two pathways were significant when tested independently under an additive disease model (Fig. 2d, FDR ≥ 0.98), or recessive or dominant models (see Methods) (Supplementary Fig. 2). Thus, the variants involved in this pathway–pathway interaction could not be identified using traditional univariate analysis or interaction tests that focus on individual SNP pairs, but were highly significant when assessed collectively by BridGE.

Furthermore, few of the pathways we discovered as implicated in a significant BPM (Fig. 3, Supplementary Data 4) would be discovered using approaches based on pathway enrichment tests of single-locus effects[3]. More specifically, Golgi-associated vesicle biogenesis, Clathrin-derived vesicle budding and the Rac-1 cell motility signaling pathway modules were enriched among the single-locus effects associated with PD at the same FDR applied to the discovery of BPMs (FDR ≤ 0.25; Supplementary Data 5). Aside from the Golgi-associated vesicle biogenesis gene set (implicated in 4 of our 23 BPMs), gene-set enrichment analysis of single-locus effects failed to identify any of the remaining 31 BPM-involved pathways (Supplementary Data 4 and 5).

The large majority (22 of 23) of discovered BPMs were associated with decreased risk for Parkinson's disease (Fig. 3). This may suggest that, in the case of Parkinson's disease, genetic interactions may be more frequently associated with protective effects, or alternatively, that there is more heterogeneity across the population in genetic interactions leading to increased risk, which would limit our ability to discover such interactions. Several BPM interactions were highly relevant to the biology of Parkinson's disease. In particular, the FcεRI signaling pathway represented a hub in the pathway-level genetic interaction network (Fig. 3). FcεRI is the high-affinity receptor for Immunoglobulin E and is the major controller of the allergic response and associated inflammation. In general, immune-related inflammation has been frequently associated with Parkinson's disease, and several immuno-modulating therapies have been pursued, but it remains unclear whether this is a causal driver of the disease or is rather a result of the neurodegeneration associated with disease progression[30,31]. There has been relatively little focus on the specific role of FcεRI in Parkinson's, but recent observations support the relevance of this pathway to the disease[32]. For example, Bower et al.[33] reported an association between the occurrence of allergic rhinitis and increased susceptibility to PD. Furthermore, reduction of IL-13, one of the cytokines activated by FcεRI and a member of the FcεRI signaling pathway, was shown to have a protective effect in mouse models of PD[34], and galectin-3, which is known to modulate the FcεRI immune response, was shown to promote microglia activation induced by α-synuclein, a cellular phenotype associated with PD[35,36]. These observations indicate that a hyperactive allergic response may predispose individuals to PD, and suggest that protective interactions discovered by our BridGE method may result from variants that subtly reduce the activity of this pathway.

Aberrant events in the Golgi and related transport processes have also been known to play an important role in the pathology of various neurodegenerative diseases, including Parkinson's disease[37,38]. For example, α-synuclein expression has been shown

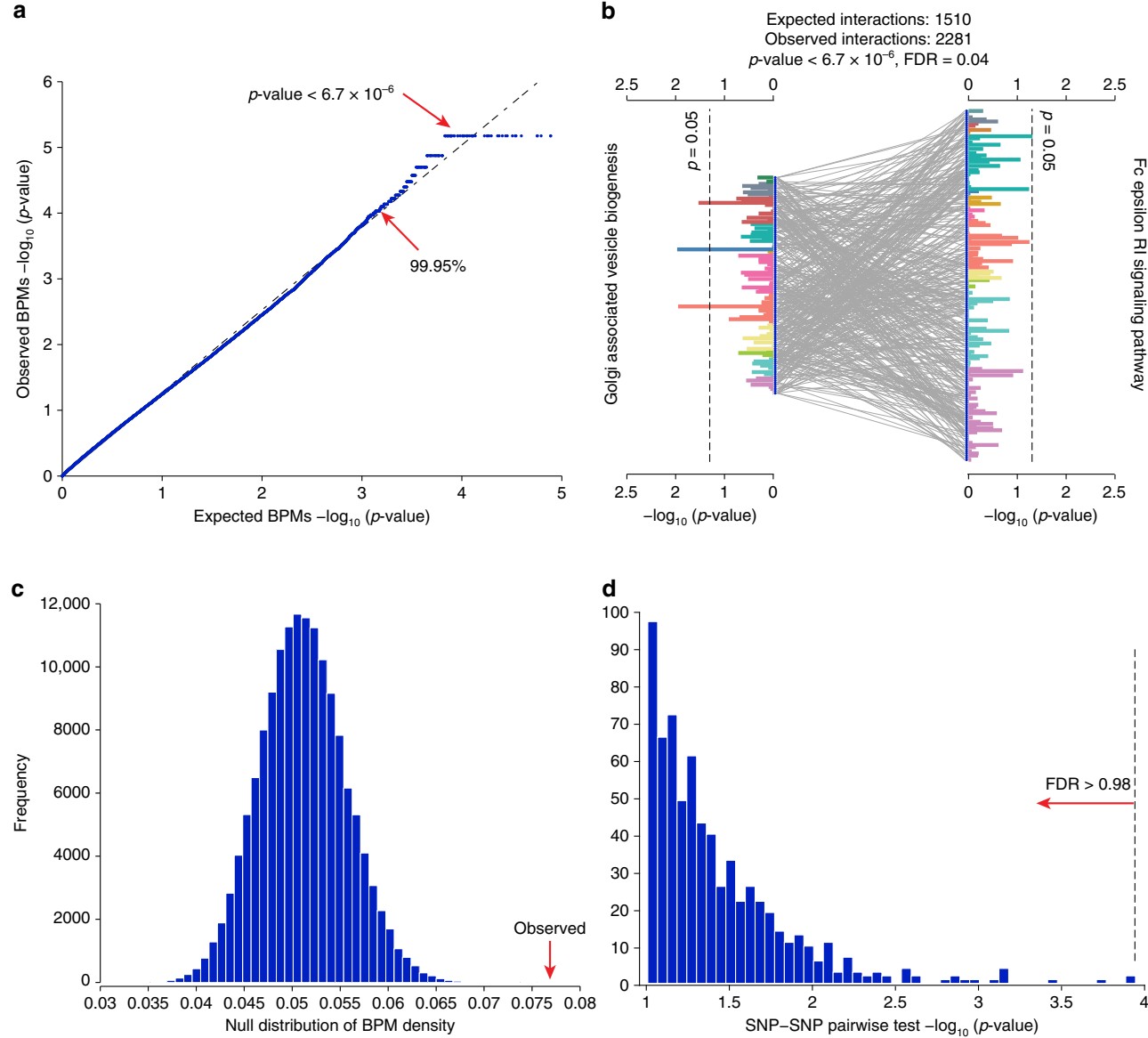

**Fig. 2** Significant pathway–pathway interactions discovered from the PD-NIA Parkinson's disease cohort. **a** Quantile–quantile (QQ) plot comparing observed $p$-values (based on SNP-pathway membership permutations) for all possible pathway–pathway interactions between the 685 pathways to the expected, uniform distribution (log10 scale). The horizontal line at $6.7 \times 10^{-6}$ reflects the maximum resolution supported by 150,000 permutations. **b** Interaction between Golgi-associated vesicle biogenesis pathway (Reactome) and FcεRI signaling pathway (KEGG). Two sets of SNPs mapped to genes in these pathways are connected by gray lines that reflect SNP–SNP interactions above a lenient top-5% percentile cutoff. The two groups of horizontal bars (grouped and colored by chromosome) show the $-\log_{10}$ $p$-values derived from a single-locus (univariate) test applied to each SNP individually (hypergeometric test), and the two dashed lines correspond to an uncorrected hypergeometric test $p \leq 0.05$ cutoff, indicating that very few of the SNPs show marginal significant association before multiple hypothesis test correction. **c** Null distribution of the SNP–SNP interaction density between the Golgi-associated vesicle biogenesis pathway and FcεRI signaling pathway described in **b** based on 150,000 SNP permutations. The observed density for the Golgi-associated vesicle biogenesis and FcεRI signaling interaction is indicated by the red arrow and was not exceeded by any of the random instances ($p_{perm} < 6.7 \times 10^{-6}$). **d** Distribution of $p$-values from individual tests for pairwise SNP–SNP interactions for SNP pairs supporting the pathway–pathway interaction, as measured by an additive disease model ($-\log_{10}$ $p$-value). None of the SNP pairs are significant after multiple hypothesis correction (dashed line at the most significant SNP–SNP pair corresponds to FDR = 0.98)

to interfere with ER-to-Golgi vesicular trafficking[38] and the overexpression of Rab1, GTPase that regulates tethering and docking of the transport vesicle with the Golgi, was shown to protect against α-synuclein-induced dopaminergic neuron loss in an animal model of PD[38]. BridGE discovered 5 distinct between- or within-pathway interactions involving the Golgi-associated vesicle biogenesis gene set, including a high confidence interaction with the FcεRI signaling pathway (Figs. 2b and 3). Interestingly, a previous study reported a genetic interaction

between *RAB7L1* and *LRRK2* in which overexpression of *RAB7L1* could rescue defects caused by *LRRK2* (ref. [39]). *RAB7L1* itself is not annotated to the Golgi vesicle biogenesis gene set, but we further investigated the individual genes driving the observed between-pathway interaction. The strongest SNP contributing to this pathway interaction was linked to the gene *RAB5C*, another Rab GTPase that regulates early endosome formation[40], for which an isoform (Rab5B) is known to physically interact with *LRRK2* (refs. [39,41]). Our result suggests that beyond this known

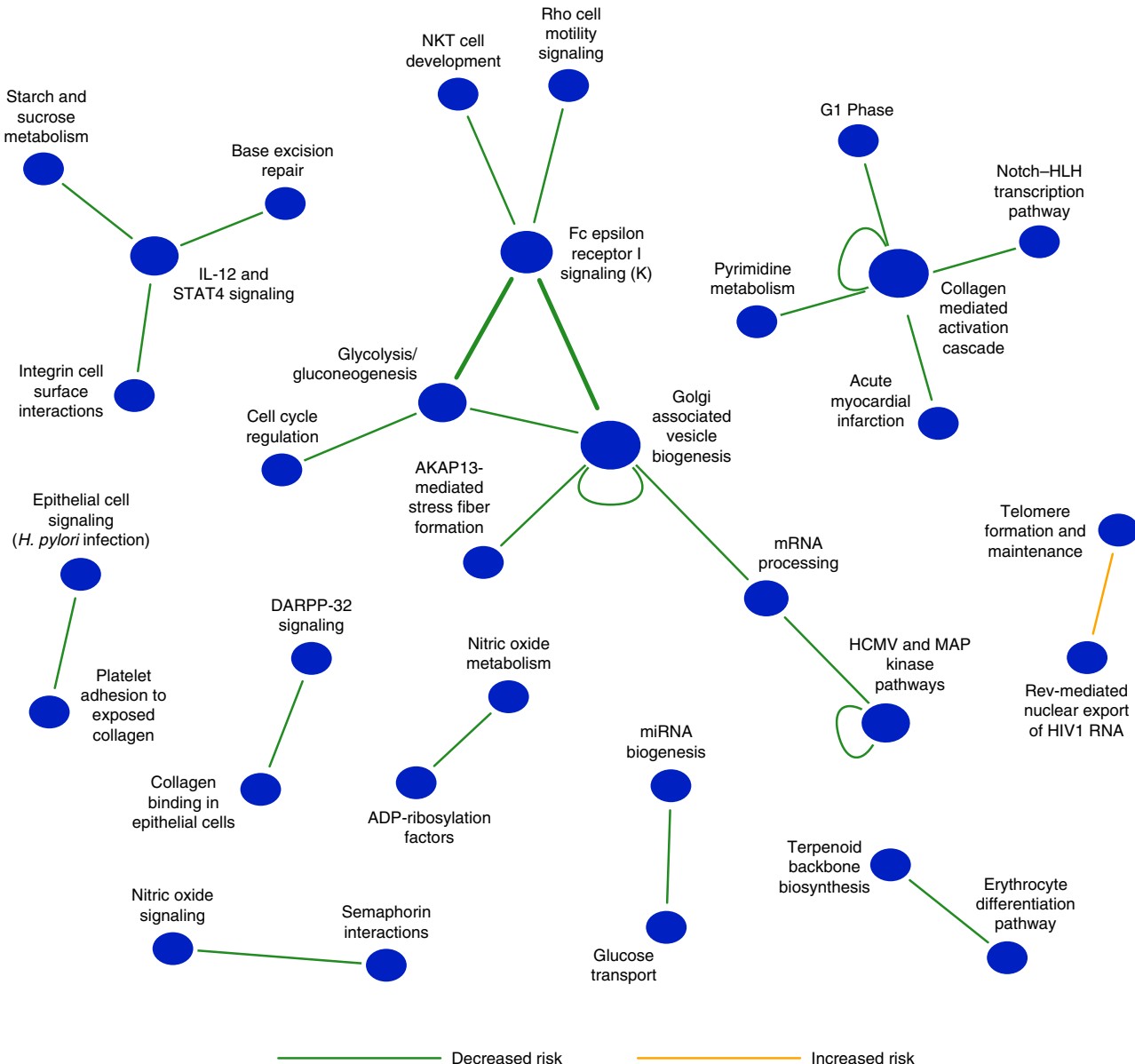

**Fig. 3** Global summary of between-pathway and within-pathway interactions discovered from a Parkinson's disease cohort (PD-NIA). Network representation of a set of significant (FDR ≤ 0.25) between-pathway (BPM) and within-pathway interactions (WPM) that are associated with increased (orange edges) or decreased (green edges) risk of PD. Each node indicates the name of the pathway or gene set, and each edge represents a between-pathway interaction or within-pathway interaction (self-loop edges). The size of the node reflects the number of interactions edges it has. Replicated interactions are shown as bold lines

interaction, there are many other combinations of common variants that affect vesicle trafficking involved in genetic interactions associated with risk of PD. BridGE also identified three protective interactions involving the IL-12 and STAT4 signaling pathway, a pro-inflammatory cytokine that has a major role in regulating both the innate and adaptive immune responses[42]. Specifically, microglial cells both produce and respond to IL-12 and IFN-gamma, and these comprise a positive feedback loop that can support stable activation of microglia[43,44], a hallmark of PD, particularly in later stages[45]. The prevalence of the FcεRI and IL-12 interactions among the discovered interactions suggests a major role for immune signaling as a causal driver of PD.

In addition to significant BPM interactions, we also discovered three significant WPMs associated with PD risk. These included interactions within the Golgi-associated vesicle biogenesis

($p_{perm} ≤ 4.7 × 10^{-5}$, and FDR < 0.01), Collagen mediated activation cascade ($p_{perm} ≤ 4.2 × 10^{-4}$, and FDR = 0.13), and the HCMV and MAP kinase pathways ($p_{perm} ≤ 2.9 × 10^{-4}$, and FDR = 0.25) (Fig. 3, Supplementary Data 4). In all three cases, minor allele combinations within the pathways were associated with decreased risk of PD. All three of these pathways were also implicated in high confidence protective BPM interactions with other pathways suggesting they play important roles in PD risk.

**Replication analysis of Parkinson's disease interactions.** To validate our findings, we tested if BPM interactions discovered in the PD-NIA cohort could be replicated in an independent PD cohort (PD-NGRC)[46] consisting of 1947 cases and 1947 controls, all of European ancestry (subjects overlapping with PD-NIA cohort were removed). At a stringent FDR < 0.05 cutoff, BridGE

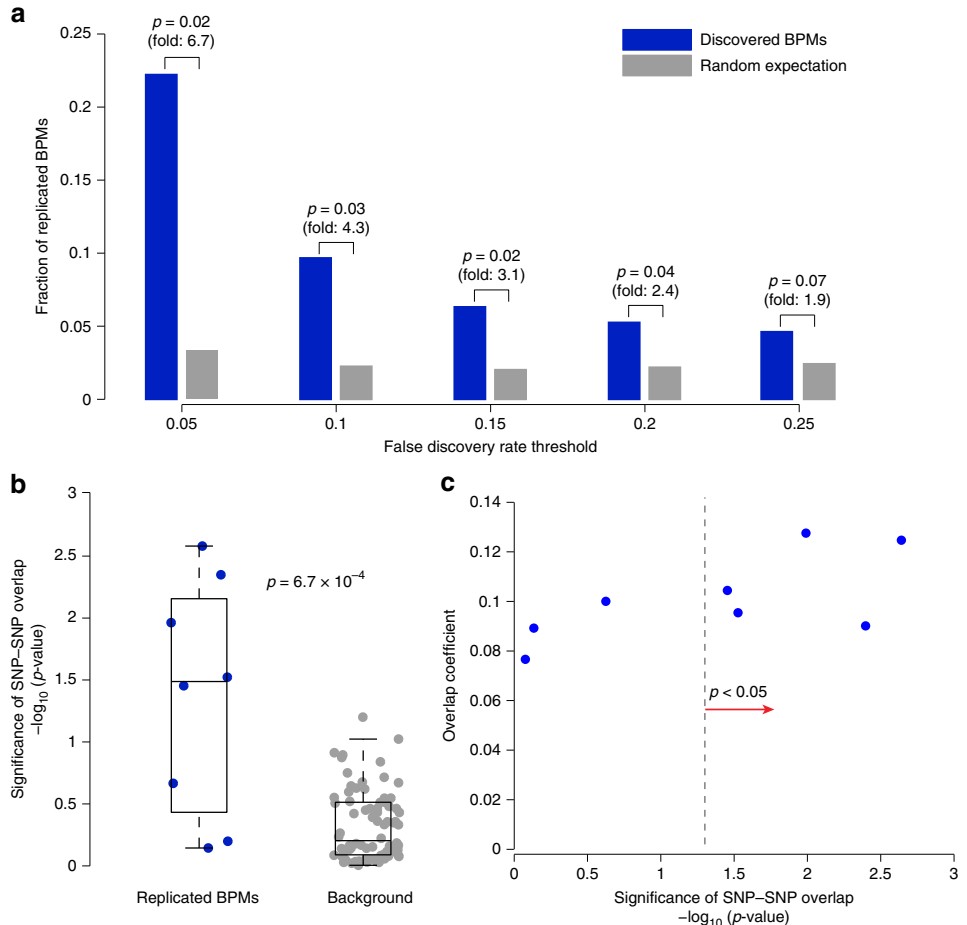

**Fig. 4** Replication analysis of BPM interactions discovered from PD-NIA in an independent cohort (PD-NGRC). **a** Each BPM interaction discovered from the PD-NIA data was tested for replication in the PD-NGRC cohort. The collective significance of replication of the entire set of interactions discovered in PD-NIA was evaluated by measuring the fraction of significant BPMs discovered from PD-NIA that replicated in the PD-NGRC cohort (blue bars) at five different FDR cutoffs (x-axis). The random expectation for the number of replicating BPMs is plotted for comparison and was estimated based on 10 random sample permutations (gray bars). **b** Sample permutation-based approach to check whether the individual SNP–SNP interactions supporting the replicated pathway-level interactions are similar between PD-NIA and PD-NGRC. The significance of the overlap (blue dots) of SNP–SNP interactions in each of the BPMs replicated in PD-NGRC was assessed by a hypergeometric test. The random expectation for the level of overlap was estimated by measuring the SNP–SNP interaction overlap in the same set of BPMs in 10 random sample permutations of the PD-NGRC cohort (gray dots). The SNP-SNP interaction overlap is modest but significantly higher than random expectation (one-tailed rank-sum test $p = 6.7 \times 10^{-4}$). **c** Scatter plot of the significance of SNP–SNP interaction overlap in each of the replicated BPMs ($-\log_{10}$ hypergeometric p-value) versus a direct measure of overlap (overlap coefficient)

discovered two BPM interactions in the PD-NIA cohort, and one of them replicated in the PD-NGRC based on all three significance criteria (permutation test $p = 0.02$, Fig. 4a) (see Methods), including the top-ranked BPM interaction we discovered between Golgi-associated vesicle biogenesis and the FcεRI signaling pathway. We further evaluated the replication rate of discovered BPMs at a range of less conservative FDR cutoffs, and indeed, we found that BPMs replicated more frequently than expected by chance and showed a stronger tendency to replicate in the independent cohort at more stringent FDR cutoffs (Fig. 4a, Supplementary Data 6). Intriguingly, another BPM interaction between the FcεRI signaling pathway and a glycolysis/gluconeogenesis gene set, also replicated providing additional evidence for the contribution of FcεRI signaling to PD risk (see Supplementary Data 6 for a complete list of replicated BPMs).

While we confirmed replication of a significant fraction of the discovered BPM interactions, this does not necessarily imply that the individual SNP pairs supporting these pathway-level effects are shared across cohorts. For the eight BPMs that were validated in the

PD-NGRC cohort, five of them exhibited significant overlap in their supporting SNP–SNP interactions, and collectively, the set of eight replicated BPMs were strongly shifted toward higher than expected SNP–SNP interaction overlap (see Methods) (one-tailed rank-sum test $p = 6.7 \times 10^{-4}$) (Fig. 4b, see Supplementary Data 6 for a list of SNP–SNP pairs in common across cohorts). However, despite statistically significant overlap among SNP–SNP interactions identified in replicated BPMs, the extent of the observed overlap in terms of fraction of pairs was relatively low for most cases, with all of them exhibiting an overlap coefficient of less than 0.15 (see Methods) (Fig. 4c). Thus, the same pathway–pathway interaction may be supported by different sets of SNP–SNP interactions in different populations, or alternatively, this may reflect that the power for reliably pinpointing specific locus pairs is limited. In either case, these results highlight the primary motivation for our method: genetic interactions, in particular those in a BPM structure, can be more efficiently detected from GWAS when discovered at a pathway or functional module level rather than at the level of individual genomic loci.

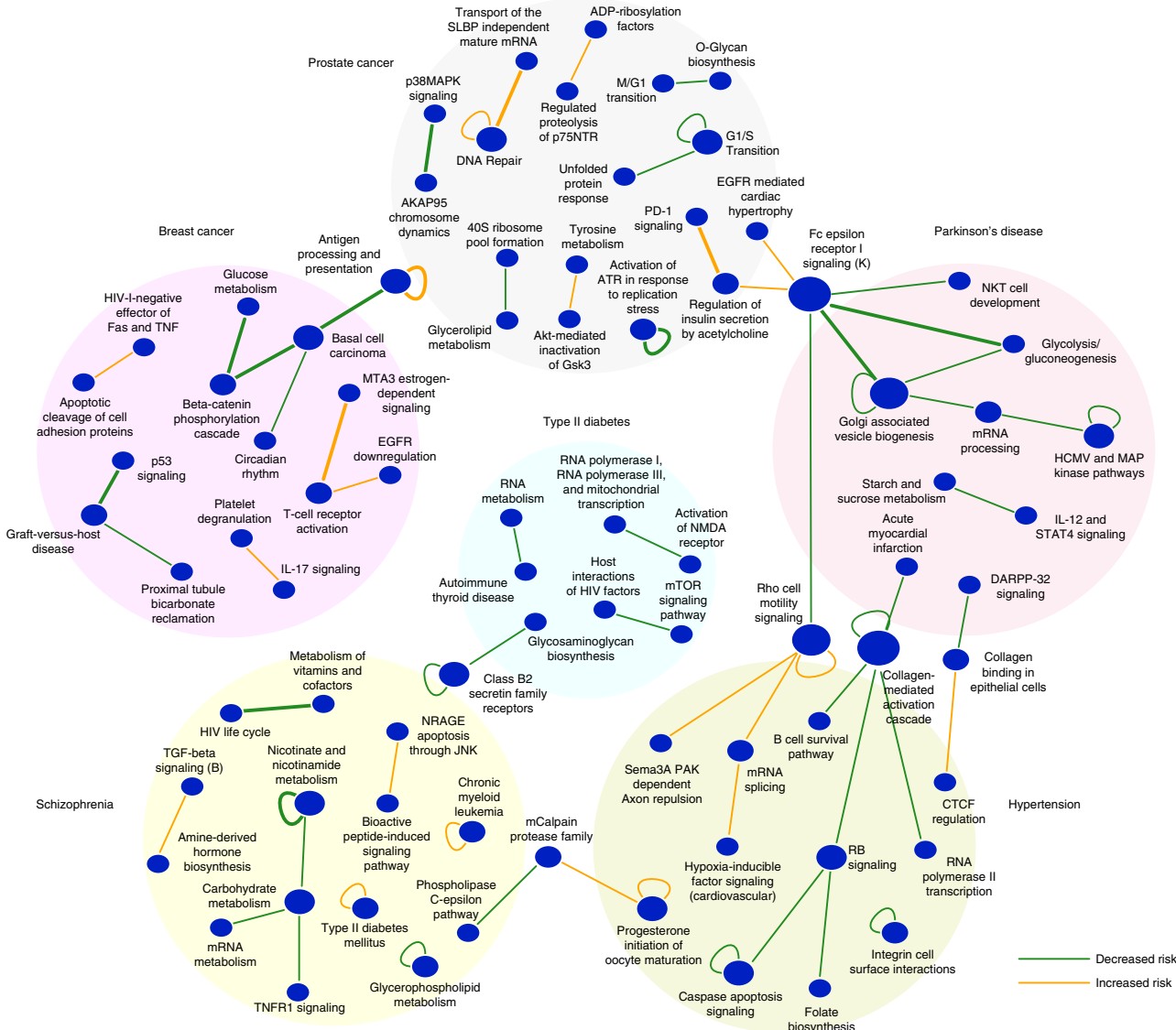

**Fig. 5** Between-pathway and within-pathway interactions discovered from six different diseases. Network representation of a set of significant between-pathway (BPM) or within-pathway (WPM) interactions (FDR ≤ 0.25) that are associated with increased (orange edges) and decreased (green edges) risk of corresponding diseases. Replicated interactions are shown as bold lines. Discoveries from different diseases are indicated by different background colors. Only the most significant 10 BPM/WPMs are shown for each GWAS cohort (see Supplementary Data 4, 9–18 for complete list)

**Discovery of genetic interactions in five other diseases.** We applied BridGE more broadly to an additional twelve GWAS cohorts representing seven different diseases (Parkinson's disease, schizophrenia, breast cancer, hypertension, prostate cancer, pancreatic cancer, and type 2 diabetes)[47–52] (Supplementary Data 1) (see Methods). Including PD-NIA, of the 13 cohorts, analysis of 11 cohorts (covering six different diseases) resulted in significant discoveries for at least one of the three types of interactions (BPM, WPM, or PATH) at FDR ≤ 0.25. More specifically, significant BPMs were discovered for eight cohorts (covering six different diseases), significant WPMs for six cohorts (covering four different diseases) and significant PATH structures for six cohorts (covering three different diseases) at FDR ≤ 0.25 (Fig. 5, Table 1, Supplementary Data 7–18). The number of interaction discoveries per cohort varied substantially, from as low as two in one of the schizophrenia cohorts to as many as 50 interactions in one of the breast cancer cohorts. While we tested multiple disease models (additive, dominant, recessive, and combined dominant–recessive), the most

significant discoveries for the majority of diseases examined were reported when using a dominant or combined model as measured by our SNP–SNP interaction metric (see Methods). The relative frequency of interactions under a dominant vs. a recessive model may be largely due to our increased power to detect interactions between SNPs with dominant effects compared to recessive effects (see Methods).

Importantly, we were able to successfully replicate genetic interactions discovered for 3 additional diseases beyond Parkinson's, including prostate cancer, breast cancer and schizophrenia (Supplementary Data 19 replication summary). For example, 3 of 9 BPMs (FDR ≤ 0.20) discovered in the ProC-CGEMS prostate cancer cohort were replicated in the ProC-BPC3 cohort (15-fold enrichment, permutation test $p = 0.0001$) while 2 of 7 WPMs discovered from the ProC-BPC3 cohort (FDR ≤ 0.10) could be replicated in ProC-CGEMS (2.22-fold enrichment, permutation test $p = 0.0001$). For breast cancer, 6 of 108 significant BPMs (FDR ≤ 0.20) discovered from the BC-MCS-JPN cohort replicated

**Table 1 Summary of discoveries across all disease cohorts**

| No. | Disease | Data cohort | Cohort size | Disease model | Structure | Min. FDR | # discovered (FDR ≤ 0.25) | # Decreased risk | # Increased risk |
|---|---|---|---|---|---|---|---|---|---|
| 1 | Parkinson's Disease | PD-NIA | 1038 | Combined | BPM | 0.04 | 23 | 22 | 1 |
| | | | | | WPM | 0 | 3 | 3 | 0 |
| | | | | | PATH | 0.1 | 3 | 3 | 0 |
| | | PD-NGRC | 3948 | Dominant | WPM | 0 | 3 | 0 | 3 |
| | | | | | PATH | 0.17 | 5 | 2 | 3 |
| 2 | Schizophrenia | SZ-GAIN | 2292 | Combined | BPM | 0.13 | 5 | 4 | 1 |
| | | | | | WPM | 0.1 | 5 | 3 | 2 |
| | | SZ-CATIE | 770 | Recessive | BPM | 0.15 | 2 | 1 | 1 |
| 3 | Breast Cancer | BC-MCS-JPN | 1384 | Dominant | BPM | 0.14 | 48 | 42 | 6 |
| | | | | | PATH | 0.05 | 2 | 2 | 0 |
| | | BC-MCS-LTN | 282 | Dominant | PATH | 0.1 | 1 | 0 | 1 |
| 4 | Hypertension | HT-eMERGE | 2450 | Dominant | BPM | 0.15 | 43 | 14 | 29 |
| | | | | | WPM | 0.1 | 3 | 1 | 2 |
| | | | | | PATH | 0 | 2 | 0 | 2 |
| | | HT-WTCCC | 3926 | Combined | WPM | 0.2 | 1 | 1 | 0 |
| | | | | | PATH | 0.15 | 2 | 2 | 0 |
| 5 | Prostate Cancer | ProC-CGEMS | 2120 | Dominant | BPM | 0.1 | 8 | 2 | 6 |
| | | ProC-BPC3 | 5474 | Dominant | BPM | 0 | 9 | 6 | 3 |
| | | | | | WPM | 0 | 5 | 2 | 3 |
| 6 | Type 2 Diabetes | T2D-WTCCC | 3854 | Combined | BPM | 0.13 | 4 | 4 | 0 |

The number of discoveries made in each of the disease cohorts evaluated, the disease model under which discoveries were made, and the direction of the disease association is reported. A complete list of interactions discovered is available as Supplementary Data 4, 9–18

in the BC-MCS-LTN cohort (twofold enrichment, permutation test $p = 0.07$) and the sole significant PATH interaction discovered from the BC-MCS-LTN cohort replicated in the BC-MCS-JPN cohort. For schizophrenia, 1 of 8 significant BPMs (FDR ≤ 0.25) discovered from the SZ-GAIN cohort replicated (fold-enrichment > 10, permutation test $p = 0.02$), and the top significant WPM (FDR ≤ 0.1) also replicated in the SZ-CATIE cohort (fold-enrichment > 10, permutation test $p = 0.03$).

The vast majority of the genetic interactions we discovered appear to be disease-specific (Fig. 5, Supplementary Data 7), and many of the pathways implicated in genetic interactions showed strong relevance to the corresponding disease. For example, we identified several cancer-related gene sets involved in replicated BPMs predicted to affect breast cancer risk, including p53 signaling, a basal cell carcinoma gene set, as well as an increased-risk interaction between *MTA3* related genes and T cell receptor activation initiated by Lck and Fyn. *MTA3* is a Mi-2/NuRD complex subunit that regulates an invasive growth pathway in breast cancer[53], and Lck and Fyn are members of the Src family of kinases whose expression have been found to be associated with breast cancer progression and response to treatment[54,55].

We also identified and replicated multiple prostate cancer risk-associated interactions that involved DNA repair, *PD-1* (programmed cell death protein 1) signaling, and insulin regulation pathways. Consistent with our findings, metabolic syndrome has been recently associated with prostate cancer[56], and serum insulin levels have been shown to correlate with risk of prostate cancer[57]. We also identified a replicating interaction associated with decreased risk of prostate cancer between the p38 MAPK signaling and AKAP95 chromosome dynamics pathways. P38 MAPK signaling has been associated with a variety of cancers[58], and *AKAP95* is an A kinase-anchoring protein involved in chromatin condensation and maintenance of condensed chromosomes during mitosis[59] whose expression has been previously implicated in the development and progression of rectal and ovarian cancers[60]. We also discovered and replicated two WPMs associated with prostate cancer risk. The first involves the antigen processing and presentation pathway (associated with increased risk) and a second involving a gene set associated with activation of ATR in response to replication stress (associated with decreased risk). Both of these pathways have strong relevance to cancer risk[61,62].

For schizophrenia, we discovered and replicated a BPM interaction comprising a gene set associated with the HIV life cycle and a vitamin and cofactor metabolism pathway. Interestingly, a recent large Danish schizophrenia study reported that schizophrenia patients are at a twofold increased risk of HIV infection, and conversely, that individuals infected with HIV exhibited increased risk of schizophrenia, especially in the year following diagnosis[63]. Our finding suggests a common genetic basis between risk factors for schizophrenia and host response to the HIV virus, which may help to explain the observed co-morbidity of these diseases. We also discovered and replicated a protective WPM for schizophrenia in the nicotinate and nicotinamide metabolism pathway. Nicotinic acid (vitamin $B_3$) supplements have been pursued as a treatment for schizophrenia dating back to the 1950s[64]. Interestingly, after an initial series of reports of promising treatments, several follow-up studies had difficulty reproducing the beneficial effects of nicotinic acid[65], which could be a result of modifier effects within this pathway. In summary, BridGE was able to detect each of these different types of pathway-level genetic interactions (BPM, WPM, and PATH) across several diverse disease cohorts, highlighting the utility of our method and the potential for genetic interactions to underlie complex human diseases.

**Power simulation study**. Several of our results indicate that the additional power gained by aggregating SNPs connecting between or within pathways is critical for discovering genetic interactions from GWAS, at least based on the cohort sizes analyzed here. To fully explore the limits of our approach, we carried out a simulation study to estimate the statistical power afforded by the BridGE method with respect to sample size, interaction effect size, minor allele frequency, and pathway size, all of which should affect the sensitivity of detection of pathway-level genetic interactions.

We focused our power analysis on the detection of BPMs, which comprise most of our discoveries. Briefly, our simulations involved two components: one in which the discovery rate of individual SNP–SNP pairs was evaluated using a simulated population cohort, and another component that simulated the detection rate of BPM interaction structures across a range of sizes given the corresponding level of false positives in the SNP-level network as determined by the first component. More specifically, for the first component of these simulations, we used GWAsimulator[66] to generate synthetic GWAS datasets with 100 embedded SNP–SNP interaction pairs under different controlled scenarios representing various combinations of different sample size, interaction effect size, and minor allele frequency. Then, we measured the discovery rate of the embedded SNP–SNP interactions. This provides a direct measure of the sensitivity and specificity of the SNP–SNP interaction-level measure that forms the basis of the pathway-level statistics. The statistics on sensitivity and specificity of SNP–SNP interaction detection from this component of our simulations were then used to guide a second set of simulations in which we assessed the sensitivity of BridGE in detecting BPMs with different levels of noise in the SNP–SNP level network. We generated a synthetic SNP–SNP interaction network matching the degree distribution of the network derived from one of the real datasets (PD-NIA, dominant–dominant model). Then, a set of non-overlapping BPMs among synthetic pathways of different sizes were embedded into this network, and BridGE was run to measure the rate of detection for each scenario (see Methods for more details on this simulation procedure).

For each parameter setting, we measured the minimum cohort size required to detect a pathway-level interaction with the corresponding properties at a fixed FDR (FDR < 0.25) (i.e., controlling the Type I error rate). Indeed, we found that each of the evaluated parameters (sample size, interaction effect size, minor allele frequency, and pathway size) affected the power of our approach (Fig. 6). As expected, the sensitivity of our method increases with increasing pathway size, which is a key motivation for the approach. For example, our power analysis indicated that a minimum cohort size of 5000 individuals (2500 cases, 2500 controls) is required to detect a 25 × 25 BPM (i.e., two interacting pathways with 25 SNPs mapping to each pathway) that confers a 2× increase in risk with a minor allele frequency (MAF) of 0.05 (FDR < 25%) while a 300 × 300 BPM with the same effect size would require only 1000 individuals (500 cases, 500 controls) for detection at the same level of significance (simulation results for more stringent FDR cutoffs). As expected, the sensitivity of the approach also increases for interactions involving SNPs with higher MAF. For example, the same 25 × 25 BPM involving variants at MAF of 0.15 conferring 2× increase in risk can be detected from cohorts as small as 2000 individuals (1000 cases, 1000 controls), and a 300 × 300 BPM with these characteristics could be detected from a cohort as small as 500 individuals (250 cases, 250 controls). A key parameter affecting these power estimates is the assumed biological density of interactions, which we define as the fraction of SNP–SNP pairs crossing two pathways of interest that actually have a functional impact on the disease phenotype relative to all possible SNP–SNP pairs. We assumed a density of 5% for the power analysis reported here (analysis based on 2.5 and 10% are included in Supplementary Fig. 3), meaning that the fraction of SNP pairs that have the potential to jointly influence the phenotype comprise only a small minority of all possible SNP pairs. In practice, we anticipate that this frequency varies substantially across different pathways, depending on the frequency of functionally deleterious SNPs that are present in the population for each pathway. A higher density of functionally deleterious SNPs will result in higher sensitivity of our approach and vice versa, a lower density of functionally deleterious SNP combinations can substantially reduce the sensitivity of our approach (Supplementary Fig. 4). Notably, while statistical power increases with pathway size (i.e., number of SNPs mapping to each pathway), this is only true under the assumption that the SNPs (and the corresponding genes) actually contribute in a functionally coherent manner to the particular pathway or functional module. On the real disease cohorts, we discovered interactions for a large range of pathway sizes (Supplementary Fig. 5), suggesting there are even relatively small functional modules (e.g., <20 associated SNPs) that have sufficiently strong interaction effects to be detected. In general, these power analyses confirm that our approach is sufficiently powered to discover pathway-level genetic interactions at moderate effect size (~2× increased/decreased risk) for relatively small cohorts (~500 or more individuals), which suggests it could be broadly applied to discover interactions in hundreds of existing GWAS cohorts that have been previously analyzed using only univariate approaches.

## Discussion

We described a novel and systematic approach for discovering human disease-specific, pathway-level genetic interactions from genome-wide association data. Genetic interactions identified from eleven GWAS cohorts representing six different diseases confirmed that structures prevalent in genetic networks of model organisms are indeed apparent in human disease populations and that these structures can be leveraged to discover significant genetic interactions that occur either between or within biological pathways or functional modules. Genetic interactions discovered for these six diseases have the potential to contribute substantially to our understanding of their genetic basis. For example, to date, there have been ~85 singly associated loci ($p \leq 1.0 \times 10^{-7}$) and one genetic interaction (between *FGF20* and *MAOB*) reported for Parkinson's Disease[67,68]. Here, we discovered 23 novel pathway-level genetic interactions, emphasizing the potential of our approach to expand our knowledge of the contribution of genetic variation associated with diseases such as PD. Indeed, many of the pathways discovered by our approach have not been previously implicated in these diseases. For example, the median percentage of BridGE-identified pathways for which there was at least one linked SNP reported in dbGaP across the six diseases was 22% (Supplementary Data 20), indicating that the large majority of our discoveries represent novel insights that could not be made using standard single-locus approaches.

There are several ways the BridGE method could be expanded and improved upon to better detect genetic interactions. First, our approach currently depends on literature-curated collections of biological pathways as a major input. The potential of our method to detect genetic interactions within or between well-defined pathways and functional modules could be substantially improved as more complete curated or data-derived functional standards are developed and integrated with the approach, which will be a focus of future work. Second, to avoid spurious network structures related to SNPs that map to genes located in close

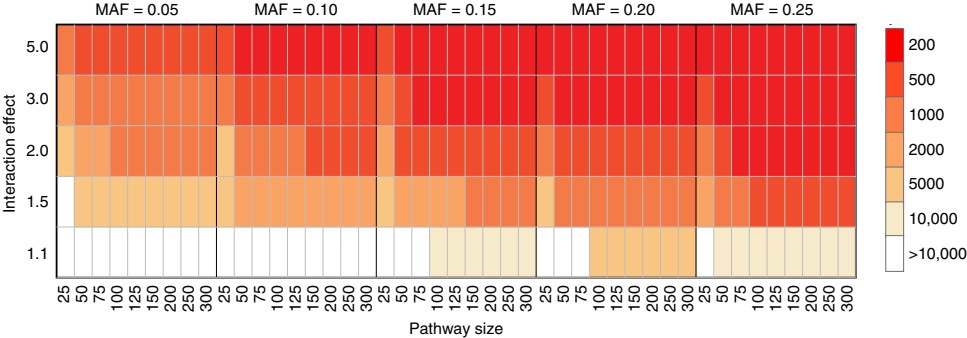

**Fig. 6** Simulation-based power analysis. Power analysis of the effect of minor allele frequency (MAF), BPM size, interaction effect size, and sample size on the discovery of between-pathway interactions. Colors in the heatmap indicate the estimated minimum number of samples needed for discovering significant BPMs of different sizes under each scenarios (MAF at 0.05, 0.1, 0.15, 0.2, 0.25; interaction effects at 1.1, 1.5, 2.0, 3.0, 5.0; BPM sizes at 25 × 25, 50 × 50, 75 × 75, 100 × 100, 125 × 125, 150 × 150, 200 × 200, 250 × 250, 300 × 300). All power analyses were conducted using a significance threshold required to meet an FDR < 0.25 (permutation test $p < 3.0 \times 10^{-5}$) based on an average of the significant BPM discoveries across all analyzed GWAS cohorts

physical proximity or LD, we sampled a conservatively sized subset of tag SNPs to run our analysis for each dataset. This conservative approach has undoubtedly missed functional variants that may contribute to disease risk. More sophisticated approaches for retaining a larger set of tag SNPs while still controlling for LD structure could improve the sensitivity of our method. Finally, we emphasize that our study focuses exclusively on detecting pathway level genetic interactions between common variants assayed by typical GWAS. Continued development to examine the contribution of rare variants or interactions between rare variants and other loci, or to leverage the full set of variants identified through whole-genome or exome sequencing represent logical extensions of the BridGE approach.

Developing mechanistic or clinically actionable disease insights based on the genetic interactions we have discovered will require additional strategies that build on pathway-level discoveries to generate more targeted hypotheses, followed by functional studies in disease models. One potential strategy to generate more targeted hypotheses involves leveraging an approach like BridGE to find pathways with robust disease-associated genetic interactions followed by a more targeted search for individual SNP–SNP or gene–gene pairs within these pathways that explain these structures. Our analysis of the Parkinson's cohort indicated that there is indeed significant overlap among the strongest SNP–SNP interactions underlying replicated pathway level interactions, supporting the potential utility of this hierarchical approach.

The extent to which genetic interactions contribute to the genetic basis of human disease has been the subject of recent debate[5,69,70]. This debate is in part fueled by differences in language among geneticists that regularly find physiological epistasis between specific alleles and statistical geneticists who measure the non-additive component of genetic variance in a population[69,71]. The target of our method is to discover disease-relevant physiological epistasis between sets of specific alleles in biological pathways based on population genetic data. Robust estimates of the additional heritability explained by pathway level genetic interactions discovered by our method will be a focus of future work, but we anticipate these genetic interactions represent just one of many contributions to heritability. Even in cases where the contribution to disease heritability is modest, genetic interactions define genetically distinct disease subtypes and point toward new insights about disease mechanism that can seed the search for new, targeted therapies. Also, recent studies suggest that accurately predicting the phenotypes of individuals from genotypes can depend critically on understanding interactions between genetic loci[69,72], and thus, progress in personalized genome

interpretation and medicine depends on our understanding of how specific alleles interact to cause phenotypes. Our work establishes a new paradigm for approaching this problem and provides a systematic method for detecting genetic interactions that can be applied to existing population genetic data for a variety of human diseases.

## Methods

**Brief summary of existing methods**. Although efficient and scalable computational tools have been developed for searching for interactions among genome-wide SNPs[10,73–75], detecting them with statistical significance remains a major challenge.

There are previous methods that have approached this problem, although from different perspectives than the method proposed here. We briefly summarize those methods and describe the novelty of our approach relative to this body of existing work.

Four general directions taken by previous methods for pathway-based analysis that are the most similar to our approach are: (1) gene-set enrichment-based approaches applied to loci derived from univariate tests, (2) gene-set enrichment-based approaches applied to SNP-level summary statistics from interactions, (3) methods that use pathways as a prior to study SNP or gene level interactions or reduce the number of hypothesis tests, and (4) regression methods for pathway-based GWAS studies.

(1) Gene-set enrichment-based approaches applied to loci derived from univariate tests: gene-set enrichment analysis (GSEA) was originally developed for case–control gene expression datasets[29,76] but has previously been adapted to summarize sets of loci (and their linked genes) derived from univariate tests applied to GWAS datasets[3]. There are two key differences between these approaches and the method we propose. First, traditional approaches for GSEA start from univariate statistics of genes or SNPs, while our approach is built on interactions between pairs of SNPs that could have little or no single-locus association with a disease phenotype. Second, approaches for GSEA target the enrichment of single gene/SNP associations in each individual pathway while our approach explores the enrichment of SNP–SNP interactions crossing each pair of pathways (between-pathway model or BPMs).

(2) Gene-set enrichment-based approaches applied to SNP-level summary statistics from interactions: The gene-set enrichment approach has also been applied beyond loci derived from univariate analysis. Another class of methods first measure genetic interactions based on pairwise SNP analysis, derive summary statistics at the individual SNP level based on specific interaction properties, and follow this with GSEA using pathway-associated SNP (or gene) interaction-based scores. For example, one such approach was recently applied to a bipolar study and a sporadic Amyotrophic Lateral Sclerosis study[17,77]. In this study, whole-genome SNPs were first filtered based on their ECML scores[78] and only the top 1000 SNPs with the strongest main effects and gene–gene interactions were retained for studying SNP–SNP interactions. Then, a SNP–SNP interaction network was constructed using a logistic regression model, and SNPs were ranked based on their network centrality in this network. Finally, candidate pathways were evaluated using a gene-set enrichment analysis based on pathway members' rankings. A similar GO enrichment approach was applied to the sporadic Amyotrophic Lateral Sclerosis study[17], but SNP interaction strength was first estimated using a multiple dimension reduction (MDR)

model and then summarized at a gene level by enrichment analysis. GO annotation enrichment approaches were then applied to these gene-level scores. Again, these studies have not introduced the key concept that motivates our method: that genetic interactions connect coherently across pairs of distinct pathways.

(3) Methods that use pathways as a prior to study SNP or gene level interactions to reduce the number of hypothesis tests: Another strategy implemented by other existing methods to address the multiple hypothesis testing challenge presented by pairwise SNP analysis is to reduce the number of hypothesis tests, based on a variety of different criteria[79]. These methods typically employ a filtering step, either data-driven[20,21,80] or knowledge-driven[81,82], before applying statistical analysis of interactions. Other illustrative examples of this class of approaches are from a recent autism spectrum disorder study where all possible SNPs were tested for interactions with the Ras/MAPK pathway[83], and a melanoma risk study where SNP–SNP interactions were studied within the five pathways that are significant based on the traditional individual SNP based-GSEA analysis[84]. Most studies implementing this approach investigate interactions among a small set of genetic variants (genes or SNPs) that either statistically demonstrate evidence for individual association with the disease phenotype or are known to be relevant to the disease based on prior knowledge. Hence, systematic detection of genetic interactions among novel genes, or genes that show no marginal association will not be detected by these approaches.

(4) Regression methods for pathway-based GWAS studies. A few regression approaches have been previously developed to leverage pathway information to model the relationship between pathway SNPs and disease phenotypes with or without considering genetic interactions, including GRASS[85], PBA[86], and EBLasso[87]. These methods typically build a regression model using genetic variants (SNPs or pairwise SNPs when considering genetic interaction) from each pathway and use ridge or lasso methods to select a subset of variants. Then, a traditional gene-set enrichment analysis or logistic regression analysis is applied on the chosen genetic variants. Such methods are conceptually different from our proposed method because they do not explicitly search for genetic interactions forming coherent within-pathway or between-pathway structures, which is key foundation of our approach. Furthermore, the methods cited above do not provide evidence for replication of the reported pathway discoveries.

In summary, existing approaches are related to the proposed approach in the general sense that they leverage existing knowledge of pathways or other sets of functionally related sets of genes to either perform enrichment on univariate effects or interaction-based SNP summary statistics (e.g., interaction degree), or simply use pathways as a prior to reduce the number of SNP pairs tested for interactions. To our knowledge, no existing methods explicitly test for higher-level interactions connecting within or between multiple pathways and are sufficiently powered to perform this systematically across comprehensive pathway databases.

## GWAS datasets

Twelve GWAS datasets, representing 13 different cohorts covering seven diseases, were used in this paper: Parkinson's disease (PD-NIA: phs000089.v3.p2, PD-NGRC: phs000196.v1.p1), breast cancer (BC-CGEMS-EUR, BC-MCS-JPN and BC-MCS-LTN: phs000517.v3.p1), schizophrenia (SCHZ-GAIN: phs000021.v3.p2; SCHZ-CATIE: CATIE study), hypertension (HT-eMERGE: phs000297.v1.p1; HT-WTCCC: cases are from EGAD00000000006, controls are from EGAD00000000001 and EGAD00000000002), prostate cancer (ProC-CGEMS: phs000207.v1.p1; ProC-BPC3: phs000812.v1.p1), pancreatic Cancer (.PanC-PanScan: phs000206.v3.p2) and Type 2 Diabetes (T2D-WTCCC: cases are from EGAD00000000009, controls are from EGAD00000000001 and EGAD00000000002). These datasets were obtained from three resources: dbGaP[88], Wellcome Trust Case Control Consortium or the National Institute of Mental Health (NIMH). Details of each dataset (e.g., sample size, genotyping platform) are summarized in Supplementary Data 1.

## Data processing

We used the same set of pre-processing steps for all GWAS datasets analyzed in this paper. Each of the steps is outlined in detail in the sections that follow.

## Sample quality control

We first controlled data quality using the standard PLINK inclusion procedure with the following parameters: 0.02 as the maximal missing genotyping rate for each individual/SNP (--mind, --geno), 0.05 as the minimum minor allele frequency (--maf), and $1.0 \times 10^{-6}$ as the Hardy–Weinberg equilibrium cutoff (--hwe 1e−6).

To identify outlier samples that were not consistent with the reported study population, we mapped SNPs in each GWAS dataset to Genome Reference Consortium GRCh37 and combined the samples with the 1000 Genomes data (all ancestry groups). We then used PLINK to perform multi-dimensional scaling (MDS) analysis. On the basis of the MDS plot, we removed samples that were not tightly clustered with the corresponding ancestry groups in the 1000 Genomes data. For the two Parkinson's disease cohorts, we followed the previous study[89] to remove samples that are likely outliers. For these cohorts, duplicate subjects were

kept in just one cohort with priority given to PD-NIA over the PD-NGRC cohort, so that we could retain as many samples as possible for the smaller cohort.

## Population stratification

*Checking relatedness among individuals*: Relatedness among each pair of subjects was tested by calculating IBD[90]. For subject pairs with a proportion IBD score greater than 0.2, one was randomly chosen and removed from the data, and the other was kept.

*Matching population structure between cases and controls*: Because spurious allelic associations can be discovered due to unknown population structure[22,91], recent GWAS analyses suggest the use of a procedure to ensure balanced population structure between cases and controls[90]. Here, all subjects were clustered into groups of size 2, each containing one case and one control that are from the same sub-population (based on pairwise identity-by-state distance and the corresponding statistical test), as is implemented in PLINK[90].

Future extensions of our method could include parameters capturing population structure directly in the model for genetic interactions, for example, as is described in ref. [92]. The primary concern in developing and applying our current approach was to ensure that population structure was not introducing spurious pathway-level interactions, so we took this relatively conservative approach to adjust for population stratification. More sophisticated approaches could reduce the number of samples lost in filtering based on population stratification and improve the sensitivity of the method.

## Filtering SNPs in LD

For each dataset, we selected all SNPs that could be mapped to at least one of the 6744 genes in the collection of pathways used in the pathway-level interaction search. A SNP was mapped to all genes that overlap with a ±50 kb window centered at the SNP, and then mapped to pathways to which the corresponding gene(s) were annotated. For the purposes of computing pathway-level statistics, a SNP was only associated once with each pathway, even if it mapped to multiple genes in the pathway.

To avoid the discovery of trivial bipartite structures, SNPs in LD need to be removed before between or within-pathway enrichment of SNP–SNP interactions is conducted. Two general approaches can be pursued towards this goal: (1) removing SNPs in LD before calculating pairwise SNP–SNP interactions; and (2) removing structures that emerge as a result of SNPs in LD after calculating pairwise SNP–SNP interactions.

The first alternative is more likely to miss informative SNP–SNP interactions than the second because it only considers a subset of all SNPs, but is more computationally efficient and scalable. It is worth noting that a biclustering algorithm pursuing the second approach was designed in ref. [16] to condense a yeast SNP–SNP interaction network into an LD-LD network. The algorithm described in that work took the SNP–SNP interaction matrix as input and searched for sets of consecutive SNPs that had a statistically significant number of across-set SNP–SNP interactions based on a hypergeometric test. The algorithm was applied on a yeast SNP–SNP interaction network (originally constructed in ref. [93]) with 1977 SNPs, where the LD effect was assumed to be localized to less than 60 SNPs for computational reasons. We attempted to apply this algorithm to the human genotype datasets used in this paper and observed that the algorithm could handle about 1500 SNPs with a threshold of $\sigma$ below 60, but not beyond. For example, on a dataset with 2000 SNPs, the program did not finish in two days with $\sigma = 100$. Given issues with scalability of this approach, we adopted the first alternative, which is to select a subset of SNPs that are not in LD.

To accomplish this, we used a procedure in PLINK[90] to select a subset of SNPs that are less likely in LD from each GWAS dataset, specifically "-indep-pairwise 50 5 0.1". With this procedure, PLINK searches each window of 50 SNPs with a sliding step of 5 SNPs, and selects a subset of SNPs with pairwise $r^2$ below 0.1 within each sliding window. After this procedure, ~15,000–20,000 SNPs were left in each dataset, and the highest $r^2$ between any pair of SNPs within any window of 1 Mb is lower than the commonly used threshold for controlling LD ($r^2 < 0.2$)[94], demonstrating that the LD was effectively controlled. Note that by using a stringent $r^2$ threshold of 0.1, we are undoubtedly ignoring many informative SNPs. However, we chose this conservative approach to minimize the chance that spurious WPMs and BPMs resulted from remaining LD structure. Future work that explores less conservative approaches to handling SNPs in LD would be worthwhile.

For diseases that we tested for replication of discovered interactions on independent cohorts of the same ancestry, to make the discovery and replication analysis consistent for these instances, cohorts were first combined and then processed using the PLINK procedures described above to select the subset of SNPs on which the analysis was run. After selection of SNPs, population stratification and discovery of interactions were then performed independently. We followed this procedure for three of the diseases analyzed, Parkinson's disease, schizophrenia, and breast cancer. For prostate cancer, our access to ProC-CGEMS and ProC-BPC3 was gained at different times, so SNPs used in ProC-BPC3 were selected based on the CGEMS cohort. A summary of all processed datasets used in this study is included in Supplementary Data 1.

## Selection of pathways

A total of 833 human pathways (gene sets) were collected from the Kyoto Encyclopedia of Genes and Genomes (KEGG)[95,96], Biocarta[24], and Reactome[25] (Supplementary Data 2). We excluded any pathway from our analysis

with less than 10 or more than 300 genes, or less than 10 or more than 300 SNPs, mapping to the pathway after LD control to avoid pathways that were too small to provide sufficient statistical power or too large to provide specific biological insights.

**SNP–SNP genetic interaction estimation**. MM, Mm and mm are used to denote the three genotypes of each SNP, i.e., majority homozygous, heterozygous, and minority homozygous, respectively. Our method implements multiple disease models, which affect how interactions are estimated at the SNP–SNP interaction level. A minor allele (m) at each locus could be additive, dominant or recessive in the context of different diseases. For the additive model, we used the standard logistic regression-based model implemented in CASSI[74] to quantify the interaction between two SNPs coded as follows, mm = 2, Mm = 1, MM = 0. In this model, the goodness-of-fit was compared between a standard logistic regression model with an interaction term between the two loci of interest and a standard logistic regression without an interaction term, and the significance of the interaction was measured by a likelihood ratio test[74]. We refer to this type of SNP–SNP interaction as an additive–additive (AA) model-based interaction. In the dominant model, a SNP is encoded as mm = 1, Mm = 1, MM = 0. In the recessive model, a SNP is encoded as mm = 1, Mm = 0, MM = 0. Because the minor allele could have recessive (R) or dominant (D) contribution to disease at two different loci comprising an interaction, four types of SNP–SNP interactions were examined: recessive–recessive (RR), dominant–dominant (DD), recessive–dominant (RD), and dominant–recessive (DR) model-based interaction for each pair of SNPs. The interactions under these four models can also be estimated by a logistic regression-based model similar to the AA case except with the appropriate encoding of the SNP genotypes. Alternatively, the RR, DD, DR, and RD interactions can be estimated by explicit statistical tests (e.g., hypergeometric tests) of the association between a specific genotype combination of two SNPs and a disease of interest, where this association is compared to the association between each of the individual SNPs and the disease (marginal effect). Interactions estimated by logistic regression-based models directly capture non-additive effects between two SNPs considering different combinations of SNP genotypes. In contrast, interactions estimated by explicit statistical tests have the flexibility of specifically testing certain combinations of genotypes for association with the phenotype. We explored alternative approaches both in representing different disease models and in the estimation of SNP–SNP interactions, and found that RR, DD, DR, and RD interactions estimated by explicit statistical tests more likely led to the discovery of significant BPMs/WPMs in the context of our BridGE approach. The measure we developed based on explicit statistical tests, called hygeSSI, is described in detail below. The relationship between hygeSSI and logistic regression-based models is explored in more depth in "Comparison with logistic regression-based interactions" section.

**hygeSSI**. We designed a hypergeometric-based measurement (hygeSSI) to estimate the interactions between two binary-coded SNPs (dominant or recessive). The hypergeometric $p$-value for a pair of binary-coded SNPs with respect to a case–control cohort is calculated as follows:

$$P_T\left(S_x, S_y, C\right) = 1 - hygecdf\left(X - 1, M, K, N\right)$$

$$= 1 - \sum_{f=0}^{X} \frac{\binom{K}{f}\binom{M-K}{N-f}}{\binom{M}{N}}$$

Where $S_x$ and $S_y$ are two SNPs; $M$ is the total number of samples; $N$ is the total number of samples in class $C$; $K$ is the total number of samples that have genotype $T$; $X$ is the total number of samples that have genotype $T$ in class $C$.

We use $P_{1\sim}(S_x, C)$ and $P_{\sim1}(S_y, C)$ to represent the individual SNP $S_x$ and $S_y$'s main effects and $P_{11}(S_x, S_y, C)$, $P_{10}(S_x, S_y, C)$, $P_{01}(S_x, S_y, C)$ and $P_{00}(S_x, S_y, C)$ to represent the effects of all pairs of combinations. With a nominal $p$-value threshold ($\alpha = 0.05$), we first require a SNP pair to have significant association with the phenotype ($P_{11}(S_x, S_y, C) \leq \alpha$). In addition, we specifically exclude instances where other allele combinations show significant association with the trait, i.e., we require: $P_{10}(S_x, S_y, C) > \alpha$, $P_{01}(S_x, S_y, C) > \alpha$ and $P_{00}(S_x, S_y, C) > \alpha$.

Given a binary-coded SNP pair ($S_x, S_y$) and a binary class label $C$, the following measure hygeSSI (Hypergeometric SNP–SNP Interaction) was defined to estimate the genetic interaction between two SNPs $S_x$ and $S_y$ (specifically for $P_{11}$):

$$\text{hygeSSI}_C\left(S_x, S_y\right) = \begin{cases} -\log_{10} \frac{P_{11}\left(S_x, S_y, C\right)}{\min\left\{P_{10}\left(S_x, S_y, C\right), P_{01}\left(S_x, S_y, C\right), P_{00}\left(S_x, S_y, C\right), P_{1\sim}\left(S_x, C\right), P_{\sim1}\left(S_y, C\right)\right\}} \\ 0; P_{11} > \alpha, P_{10}, P_{01}, P_{00} \leq \alpha \end{cases}$$

As described in a recent comprehensive review[7], algorithms based on logistic/linear regression, multifactor dimensionality reduction (MDR)[97], entropy or information theory[98] have been developed to measure genetic interactions. All of these approaches quantify the synergistic effect of SNP pairs by comparing the relative strength of the association between a pair of SNPs and a disease trait with the strength of the associations between two individual SNPs and the disease trait. A few of these alternatives were tested in the context of our method and did not

provide the significant results we achieved with the metric above. We designed the above hygeSSI measure because it explicitly captures the interaction between combinations of specific genotypes of two loci.

**Construction of SNP–SNP interaction networks**. We constructed SNP–SNP interaction networks to serve as the basis for the pathway-level interaction tests based on each of the disease model assumptions. An AA interaction network was constructed by the described logistic regression-based approach, where SNP–SNP edge scores were derived from the $-\log_{10} p$-value resulting from the likelihood ratio test. The RR and DD interaction networks were computed based on the hygeSSI metric, and only positive interactions were kept in the network (i.e., where the joint effect of the SNP–SNP pair under the corresponding disease model was stronger than any marginal or alternative combination of SNPs). In addition to the above three networks, we also constructed a hybrid SNP–SNP interaction network in which interactions under recessive and dominant disease model could coexist. To do this, we integrated all four networks (RR, DD, RD, and DR) into a single network (RD-combined) by taking the maximum hygeSSI among the four interaction networks for each pair of SNPs.

**Measuring pathway–pathway interactions**. For each pair of pathways, we want to test if the number of SNP–SNP interactions between them is significantly higher than expected given the overall density of the SNP–SNP network as well as the marginal interaction density of the two pathways involved. Enrichment analysis based on SNP–SNP interactions is much more computationally challenging, and thus we choose to binarize the hygeSSI values (based on a lenient threshold) to make follow-up computation efficient and scalable. After binarization, we divided the SNP–SNP interaction network into two networks based whether the joint mutation of a SNP pair is more prevalent in the case or control group, which we refer to as the risk and protective networks, respectively.

For each pathway–pathway interaction, we first removed the common SNPs shared between two pathways. Then, we test if the observed SNP–SNP interaction density between two pathways is significantly higher than expected globally (the global network density) and locally (the marginal density of SNP–SNP interactions of the two pathways). Specifically, the marginal density of a pathway is calculated as the SNP–SNP interaction density between the SNPs mapped to the pathway and all other SNPs in the network. We computed a chi-square statistic to test differences from both global and local density, namely chi-square global ($X_{global}^2$) and chi-square local ($X_{local}^2$). The chi-square test assumes the SNP–SNP interactions in a network are independent, which may not be true for a variety of reasons. So, in addition to these chi-square statistics, we use permutation tests to derive an empirical $p$-value for each pathway–pathway interaction. To do this, we randomly shuffled the SNP-pathway membership (NP = 100,000–200,000 times), and for a given pathway–pathway interaction (bpm$_i$), we compared its observed $X_{global}^2$ and $X_{local}^2$ with the values from these random permutations ($\tilde{X}_{global}^2$ and $\tilde{X}_{local}^2$) to obtain a permutation-based $p$-value.

$$p_{perm}\left(bpm_i\right) = \frac{\#\left(\tilde{X}_{global}^2 \geq X_{global}^2\left(bpm_i\right) \& \tilde{X}_{local}^2 \geq X_{local}^2\left(bpm_i\right)\right) + 1}{NP}$$

We used ($p_{perm}$) together with ($X_{global}^2$) and ($X_{local}^2$) for BPM discovery as further described in detail in the next two sections.

**Correction for multiple hypothesis testing**. Because a large number of pathway pairs (all possible pathway–pathway combinations) are tested in the search for significant BPMs, correction for multiple hypothesis testing is needed. To estimate a false discovery rate, we employed sample permutations (NP = 10 times) to derive the number of expected BPMS discovered by chance at each level of significance. We randomly shuffled the original case–control groups 10 times while maintaining the matched case–control population structure. For each permuted dataset, the same, complete pipeline for BPM discovery was performed, including calculation of the SNP–SNP interaction network after permutation, which was then thresholded at a fixed interaction density matching the density chosen for the real sample labels. From these sample permutations, we obtained three null distributions ($\tilde{X}_{global}^2$, $\tilde{X}_{local}^2$, and $\tilde{p}_{perm}$), from which we estimated the FDR for each BPM (e.g., bpm$_i$). Specifically, we compared the number of BPMs observed in each real dataset that have better overall statistics than $bpm_i$ with the corresponding random expectation estimated from the three null distributions derived from sample permutations ($\tilde{X}_{global}^2$, $\tilde{X}_{local}^2$, and $\tilde{p}_{perm}$):

$$\text{FDR}\left(bpm_i\right) = \frac{\#\left\{\tilde{X}_{global}^2 > X_{global}^2\left(bpm_i\right) \& \tilde{X}_{local}^2 > X_{local}^2\left(bpm_i\right) \& \tilde{p}_{perm} < p_{perm}\left(bpm_i\right)\right\} / NP}{\#\left\{X_{global}^2 > X_{global}^2\left(bpm_i\right) \& X_{local}^2 > X_{local}^2\left(bpm_i\right) \& p_{perm} < p_{perm}\left(bpm_i\right)\right\}}$$

A simpler approach to estimate FDR would be to use only the SNP permutation-based $p$-value, $p_{perm}$, in the above formula. However, we chose to use all three measurements ($X_{global}^2$, $X_{local}^2$, and $p_{perm}$) because we observed that in some cases the permutation-based $p$-value alone did not provide enough resolution to differentiate among top BPMs (this could be improved with additional SNP permutations, but this is computationally expensive). $X_{global}^2$ and $X_{local}^2$ provide higher resolution

measures of significance of each BPM and, when combined with the permutation-based p-value, can differentiate among the top-most significant discoveries.

We emphasize that we have used a hybrid permutation strategy to assess significance of the discovered structures. The primary permutation applied was to permute the SNP labels, for which 100,000–200,000 permutations were used for each dataset analyzed. The sample (case–control label) permutation approach mentioned above was used in addition to the SNP permutation strategy to estimate our false discovery rate across all discovered interactions. For each of the 10 sample permutations, we ran the full set of 100,000–200,000 SNP permutations. This hybrid approach provides a robust estimate of significance of the discovered pathway interactions and properly corrects for multiple testing.

We also conducted a study to explore the sensitivity of our FDR estimation on the number of sample permutations. Specifically, for the PD-NIA dataset, we performed 1000 sample permutations (and 200,000 SNP permutations within each of these) to derive an estimate of FDR for discoveries in this dataset (Supplementary Data 23). As shown in Supplementary Fig. 6, the FDRs estimated from 10 sample permutations show reasonable agreement to FDRs estimated from 1000 sample permutations (Pearson's correlation of 0.81).

**Selection of disease models and density thresholds**. The method we proposed for pathway-level detection of genetic interactions is general in the sense that any disease model (e.g., RR, DD, RD-combined, and AA) or interaction statistic could be used to discover pathway-level interactions. In this study, we focus on prioritizing a single disease model per disease cohort for full analysis by our pipeline to limit the complexity of data analysis across the 13 GWAS cohorts we explored with our method. Here, we describe the strategy we used to select the disease model to focus on for each GWAS dataset.

To prioritize the disease model and SNP–SNP interaction network density threshold for each dataset, we first performed a pilot experiment in which we examined combinations of different disease models and different density thresholds, but with fewer SNP permutations (Supplementary Data 21). To exclude SNP pairs with little or weak interactions from our analysis, we required each SNP pair's hygeSSI score to be at least 0.2 before applying density-based binarization. For each combination, we performed 10,000 SNP-pathway membership permutations (as compared to 100,000–200,000 for a complete run) to estimate FDRs using a similar procedure as that described in the section "Correction for multiple hypothesis testing", except that SNP permutations were used to estimate FDR instead of sample permutations, as sample permutations are much more computationally expensive. On the basis of this pilot experiment in each cohort, we chose the disease model and density threshold combination that resulted in the lowest estimated FDR for the top-most significant pathway–pathway interaction. The rationale for using such a pilot experiment is to identify the disease model that is most likely to discover significant pathway-level interactions while limiting the computational burden of applying our approach to several GWAS cohorts under multiple disease models. Based on these pilot experiments, which were performed for all 13 cohorts, we ran the complete BridGE pipeline, including 100,000–200,000 SNP permutations and 10 sample permutations with the disease model and network density threshold chosen from the pilot experiments. The results of pilot experiments for all cohorts are reported in Supplementary Data 21, and all full BPM discovery results for all diseases can be found in Supplementary Data 3 and 9–18 as well as a summary in Supplementary Data 8. We note that for focused application of our approach on a single or small number of cohorts of interest, we would suggest exploring all possible disease models with complete runs.

**Replication in independent cohorts**. The significant BPMs discovered from one cohort could be evaluated in another independent cohort for replication. To determine whether a discovered BPM was replicated in an independent cohort, we required the BPM to satisfy $X^2_{global}$ test $p \leq 0.05$, $X^2_{local}$ test $p \leq 0.05$, and $p_{perm} \leq 0.05$ on the validation cohort. We also performed sample permutation tests (NP = 10) for each validation cohort, from which we could generate null distributions for $X^2_{global}$, $X^2_{local}$, and $p_{perm}$ in the validation cohort. Given a set of discovered BPMs (e.g., FDR $\leq$ 0.25), we calculated fold enrichment by comparing the number of BPMs discovered from the original dataset that passed the validation criteria to the average number of BPMs that passed the same validation criteria in the random sample permutations. More specifically, given a set of significant BPMs (bpm$_{1,2,\ldots,k}$) which were discovered from original cohort, the fold enrichment for replication is defined as:

$$\text{Fold} = \frac{\#\left\{p(X^2_{global}) \leq 0.05 \& p(X^2_{local}) \leq 0.05 \& p_{perm} \leq 0.05\right\}}{\#\left\{p(\tilde{X}^2_{global}) \leq 0.05 \& p(\tilde{X}^2_{local}) \leq 0.05 \& \tilde{p}_{perm} \leq 0.05\right\}/\text{NP}}, \forall \text{bpm}_{1,2,\ldots,k}$$

We also evaluated the significance of the fold enrichment by 10,000 bootstrapped BPM sets. Specifically, we randomly selected the same number of BPMs and used the above procedure to evaluate the fold enrichment, and we repeated this for 10,000 times to generate a null distribution for the fold enrichment scores in the validation cohort. We then evaluated the significance of the fold enrichment score for our discovered BPM set based on this empirical null distribution. All replication results can be found in Supplementary Data 6 and 19.

For the BPMs that replicated in an independent cohort, we further checked if the SNP–SNP interactions supporting the discovered pathway-level interactions were similar between the cohort used for discovery and the independent cohort

used for replication. For example, we used the BPMs discovered from PD-NIA (FDR $\leq$ 0.25) and for each BPM replicated in PD-NGRC, we computed the number of SNP–SNP interactions in common between the PD-NIA and PD-NGRC interaction networks as supporting interactions for the BPM. We used the same permutation approach as that described in the "Correction for multiple hypothesis testing" section for BPM-level validation except that the SNP–SNP interactions supporting each BPM were compared between the discovery and validation cohorts by a hypergeometric test. This was done for the real validation cohort PD-NGRC first and then repeated 10 times under sample permutations of the validation cohort to estimate a null distribution. A Wilcoxon's rank-sum test was then used to evaluate the significance of the SNP–SNP interaction overlap between the replicated BPMs in the real validation cohort and in the random sample permuted validation cohorts (Fig. 4b).

**BPM redundancy**. Due to the fact that many of the curated gene sets overlap, we needed to control for redundancy in the discovered BPMs. To do this, in reporting total discoveries, we filtered BPMs based on their relative overlap in terms of SNP–SNP interactions using an overlap coefficient. The overlap coefficient between two BPMs is defined as the number of overlapping SNP pairs divided by the number of possible SNP pairs in the smaller BPM.

For the significant BPMs discovered, we computed all pairwise overlap coefficients and used a maximum allowed similarity score of 0.25 as a cutoff. We reported the number of unique BPMs based on the number of connected components. For visualization purposes (Fig. 3), we selected representative BPMs from each connected component, prioritizing BPMs that validated in the independent cohort (PD-NGRC) for visualization. Significance of the validation of the set of BPMs was evaluated on the entire set of discovered BPMs using the permutation procedures described above, which directly accounts for the redundancy among the discovered BPMs.

**Measuring within-pathway interactions**. In addition to the between-pathway model (BPM), we also tested for enrichment of genetic interactions within each pathway[14] (within-pathway models, WPMs). All of the measures and procedures described above for BPMs apply directly to the WPM case, only we specifically look at SNP pairs connecting genes within the same pathways/gene sets instead of between-pathway pairs. For WPMs, the false discovery rate and validation statistics were computed separately from BPMs. All WPM discovery results can be found in Supplementary Data 3, 9–18.

**Identifying pathway hubs in the SNP–SNP interaction network**. Since both "between-pathway model" and "within-pathway model" analysis have been designed to avoid discoveries caused by the higher marginal interaction density of the individual pathways, pathways that are frequently interacting with many loci across the genome (as opposed to localized interactions with functionally coherent gene sets) are less likely to appear in our pathway–pathway or within-pathway interactions. However, such pathways may also be disease relevant as they reflect pathways that modify the disease risk associated with a large number of other variants, so we also report pathways exhibiting these characteristics with BridGE (we refer to these as "PATH" discoveries in BridGE output files). For PATH discovery, the procedure is similar to that for BPMs and WPMs, with a minor modification to the scoring of each pathway. Specifically, each pathway is represented by a vector of pathway-associated SNPs' degrees ($D_{path}$) in the SNP–SNP interaction network. We then applied a one-tailed rank-sum test ($U_{path}$) to compare each pathway-associated degree vector with the non-pathway-associated degree vector ($D_{non\_path}$) to see if the PATH-associated SNPs exhibited significantly more interactions than the entire set of SNPs. PATH discovery and validation is then done by repeating the same steps as BPM/WPM discovery but replacing the $X^2_{global}$ and $X^2_{local}$ statistics with the rank-sum test $U_{path}$ p-value (in $-\log_{10}$ scale). All PATH discovery results can also be found in Supplementary Data 3 and 9–18. Many of these also have clear relevance to the disease cohort in which they were discovered. For example, applying BridGE to discover such hub pathways in the context of Parkinson's disease resulted in three significant pathways after removing redundancy (FDR $\leq$ 0.25), including the same Golgi-associated vesicle biogenesis gene set as well as the IL-12 and STAT4 signaling pathway (Biocarta) discussed in the main text.

**Comparison with logistic regression-based interactions**. We examined if the interactions captured by hygeSSI were non-additive as measured through a standard logistic regression-based interaction measure. We applied the logistic regression model on the PD-NIA data and computed RR, DD, RD, and DR interaction networks (binary encoding as described earlier). We also integrated these four logistic regression-based networks to form an RD-combined network. Then we checked (1) if the top SNP–SNP interactions based on hygeSSI were significant ($p \leq 0.05$) in logistic regression-based tests, and (2) if the significant BPMs discovered from a hygeSSI interaction network show significance ($p(X^2_{global}) \leq 0.05$, $p(X^2_{local}) \leq 0.05$, and $p_{perm} \leq 0.05$) based on SNP–SNP interactions estimated from logistic regression. This analysis revealed that among the top 1% hygeSSI interactions, 93% are significant based on a logistic regression-based test for interaction. And for the significant BPMs (FDR $\leq$ 0.05), 100% of them are

also significant if only SNP–SNP interactions also supported by a logistic regression model are considered. These data suggest SNP–SNP interactions captured by hygeSSI do represent non-additive interactions as defined based on a logistic regression model. Detailed results from this comparison can be found in Supplementary Data 22. Further evaluation of different disease models and different measures for estimating SNP–SNP interactions in the context of BridGE will be the focus of future work.

**Evaluation of significance of SNP–SNP interaction tests**. For SNP–SNP pairs that supported the between-pathway interaction reported in Fig. 2b, we checked the statistical significance of SNP–SNP interaction pairs tested individually. We measured all pairwise AA, RR, DD interactions. We then performed a permutation test in which sample labels were permuted 10 times and for each permutation, all pairwise AA, RR, DD interactions were computed for each SNP pair. These permutations were used to estimate a FDR for those SNP–SNP pairs supporting the reported BPM. No individual SNP–SNP pairs were significant after FDR-based multiple hypothesis correction (Fig. 2d, Supplementary Fig 2).

**Pathway enrichment analysis of single-locus effects**. To check if the pathways involved in the significant BPMs discovered in PD-NIA were enriched for SNPs with moderate univariate association with Parkinson's disease, we performed single pathway enrichment analysis for the same set of 685 pathways used for BPM discovery. In the single pathway enrichment analysis, we used a hypergeometric test as the SNP-level statistic for measuring univariate association (risk and protective associations were evaluated separately) for three different disease models: (1) recessive; (2) dominant; and (3) a combination of recessive and dominant, in which each SNP were tested for both recessive and dominant disease models and the more significant one assigned to each SNP. We then used Wilcoxon's rank-sum test to check if a pathway was enriched for SNPs with higher association than the background (all SNPs). With 10,000 sample permutations, we computed FDR for each individual pathway (both risk and protective associations) by using same procedure described in "Correction for multiple hypothesis testing" section. The results are summarized in Supplementary Data 5.

**Comparison of BridGE discoveries with GWAS catalog results**. To check if previous singly associated SNPs also appear in our discovered pathway-level interactions, we compared our BridGE-discovered pathways with pathways that could be linked to disease risk loci reported in NHGRI-EBI GWAS catalog[67] (Ensembl release version 87, retrieved on Feb 6, 2017). Based on the GWAS catalog, the numbers of genes linked to known risk loci ($p \leq 2.0 \times 10^{-5}$) in each disease are: 143 (144 SNPs, Parkinson's disease), 1009 (824 SNPs, Schizophrenia), 134 (172 SNPs, Breast cancer), 71 (57 SNPs, Hypertension), 249 (234 SNPs, Prostate cancer), and 294 (288 SNPs, Type II diabetes). For each disease, we summarized all pathways that were discovered by BridGE (FDR ≤ 0.25) and identified pathways that were implicated by individually associated SNPs reported in the GWAS catalog (a SNP mapping to a single gene in a given pathway was assumed to implicate the corresponding pathway). For context, for each disease, we also summarize the total number of genes implicated by GWAS-identified SNPs, how many these map to the 833 pathways we used in our study, and how many of them can be linked to the significant pathways identified by BridGE. These results are presented in Supplementary Data 20.

**Dependence of interaction discoveries on disease model**. While we tested multiple disease models (additive, dominant, recessive, and combined dominant–recessive), the most significant discoveries for the majority of diseases examined were reported when using a dominant or combined model as measured by our SNP–SNP interaction metric. The relative frequency of interactions under a dominant vs. a recessive model may be largely due to our increased power to detect interactions between SNPs with dominant effects compared to recessive effects. More specifically, individuals with both heterozygous and homozygous (minor allele) genotypes at two interacting loci would be affected under a dominant disease model, while only individuals with homozygous (minor allele) genotypes would be affected in a recessive disease model. The number of individuals homozygous at two interacting loci can be quite small depending on the allele frequency, which limits our power to discover them. Thus, the larger number of discoveries based on a dominant model assumption relative to a recessive model is likely a reflection of difference in statistical power and not an indication that genetic interactions among alleles with dominant effects are contributing more strongly to disease risk. We observed that interactions derived from an additive disease model provided the fewest significant discoveries when used in the context of BridGE based on the pilot experiments (Supplementary Data 21). To understand this, we investigated whether the SNP–SNP interactions supporting the BPMs discovered under the combined dominant–recessive model for the PD-NIA cohort were non-additive when evaluated using a logistic regression-based interaction test as opposed to the direct association tests used for our dominant and recessive disease models. Most SNP–SNP interactions supporting the PD-NIA discoveries were indeed non-additive when assessed using the logistic regression framework, but these were not necessarily ranked among the highest SNP–SNP pairs when assessed in the context of a logistic regression model (Supplementary Data 22), which may explain the

difference in results under the additive vs. recessive or dominant disease models. An important distinction between the SNP-level interaction metric we use is that we specifically identify the small subset of individuals with the appropriate combination of genotypes (dominant model: heterozygous for minor allele at two candidate loci; recessive model: homozygous for minor allele at two candidate loci), and directly test for association with the disease phenotype, whereas for the additive model, an interaction term must explain a sufficient fraction of the variance across the entire population for it to reach significance. This distinction may play a role in why we are able to discover pathway-level genetic interactions with the metric proposed here but rarely with a standard additive model. It is worth noting that the core of the BridGE approach, discovering genetic interactions in aggregate rather than in isolation, is readily adaptable to other disease models or other statistical measures of interaction. Further exploration of different disease models as well as different statistical measures of interaction[93,99] would be worthwhile.

**Power analysis based on interaction simulation study**. To characterize the power of our BridGE approach with respect to sample size, effect size, minor allele frequency and pathway size, we used a two-stage simulation approach. We first generated synthetic GWAS datasets with embedded SNP–SNP interaction pairs using GWAsimulator[66]. Specially, we used PD-NIA as input to GWAsimulator and embedded SNP–SNP interactions with different minor allele frequencies (e.g., 0.05, 0.1, 0.15, 0.2 and 0.25) and a range of interaction effects (e.g., $d_{11} = d_{12} = d_{12} = d_{22}$ = 1.1, 1.5, 2, 2.5, 3, and 5, where 0, 1, 2 refer to the number of minor alleles present in a given genotype for an individual SNP, and $d_{11}$, $d_{12}$, $d_{12}$, and $d_{22}$ are defined as the relative risk of that genotype--11,12, 21 or 22-- versus 00)[66]. We also varied the number of samples (genotypes) in the simulation (e.g., 200, 500, 1000, 2000, 5000, and 10,000). In all simulations, we specified the disease prevalence to be 0.05, dominance effect for all disease SNPs with PR1 = 1 (see GWAsimulator for more details)[66]. Under different scenarios (combinations of different minor allele frequencies, interaction effects and sample sizes), we embedded 100 SNP pairs and measured the percentage of SNP–SNP interactions that were identified by our pairwise SNP–SNP interaction measure, hygeSSI at a 1% network density (e.g., SNP–SNP pairs whose hygeSSI is greater or equal to the 99th percentile of all possible interactions) (Supplementary Fig. 7). These simulations provide a direct measure of the sensitivity and specificity of the SNP–SNP interaction level measure that forms the basis of the pathway-level power statistics.

The SNP–SNP level power statistics (sensitivity) were complemented with a second set of simulations in which we directly assessed the sensitivity of BridGE in detecting BPMs with different levels of noise in the SNP–SNP level network (derived from the process described above). To characterize the statistical power of our approach as a function of pathway size, we first generated a synthetic interaction network with the same degree distribution as the PD-NIA DD network at 1% density. Then, we embedded a set of non-overlapping BPMs into this SNP–SNP interaction network while retaining the same degree distribution and density of the network. Each set had 90 BPMs at 9 different sizes (number of SNPs mapped to the two pathways in each BPM: 10 × 10, 25 × 25, 50 × 50, 75 × 75, 100 × 100, 150 × 150, 200 × 200, 250 × 250 and 300 × 300); and 10 different background densities 0.01, 0.012, 0.014, 0.016, 0.018, 0.02, 0.025, 0.03, 0.04, and 0.05. We applied 150,000 SNP-pathway membership permutations to assess the significance of these embedded patterns. The SNP permutation-derived p-values of the simulations were reported in Supplementary Fig. 4 and provide an estimation of BPM density required for detecting interactions between pathways of different sizes. We used the average p-values (SNP permutation $p = 3.0 \times 10^{-5}$) of the significant BPM discoveries across all GWAS cohorts (FDR ≤ 0.25) as the discovery significance cutoff for the simulation analysis.

We derived power estimates for each combination of parameter settings by integrating the results from the two simulation studies above. More specifically, we estimated the minimum sample size needed to discover significant BPMs at different pathway sizes under each of the scenarios (e.g., minor allele frequency, relative disease risk). To connect the two simulation studies, we require a scaling parameter (here, we explored $s = 0.025$, 0.05, and 0.1) which corresponds to the biological density of genetic interactions crossing each pair of truly interacting pathways. This represents the fraction of all possible SNP–SNP pairs crossing the pair of pathways of interest for which the combination of variants actually has a functional deleterious impact on the phenotype. This quantity is expected to be relatively small, but is difficult to estimate, which is why we have explored three scenarios ($s = 0.025$, 0.05 and 0.1). For a given BPM of a specific size (10 × 10, 25 × 25, 50 × 50, 75 × 75, 100 × 100, 150 × 150, 200 × 200, 250 × 250 and 300 × 300), from the 2nd simulation, we identified the corresponding BPM density (Density$_{BPM}$) needed for it to rise to the level of statistical significance required for a 25% FDR based on the PD-NIA cohort. We then scaled the required density by the parameter, s, and based on the 1st set of simulation results, identified the minimum sample size required under each scenario (combinations of minor allele frequency, interaction effect, and sample size) to support the discovery of the corresponding BPM: Sensitivity × s ≥ Density$_{bpm}$. Results are summarized in Fig. 6.

Simulation results for additional scaling parameters ($s = 0.1$ and $s = 0.025$) are included in the Supplementary Fig. 3. These plots together provide an estimate of the power of the BridGE approach to detect pathway–pathway interaction in these different scenarios. We note that this power analysis was conducted for the

dominant disease model, which comprises the majority of the BPM interactions discovered across all cohorts. Sensitivity of our method under a recessive model assumption is expected to be lower, which is consistent with the relative rate of discoveries of both types. We note that our power analysis accounts for both Type I and Type II error. Specifically, the simulations directly reflect the sensitivity (sensitivity = [1 – Type II error]) as a function of sample size in discovering BPMs under the practical scenario where Type I error rate is controlled (through control of the FDR) for exhaustive pairwise tests for BPMs.

**Reporting Summary**. Further information on research design is available in the Nature Research Reporting Summary linked to this article.

## Data availability

The genome-wide association datasets (PD-NIA: phs000089.v3.p2, PD-NGRC: phs000196.v3.p1, SZ-GAIN: phs000021.v3.p2, BC-CGEMS-EUR: phs000147.v3.p1, BC-MCS-JPN: phs000517.v3.p1, BC-MCS-LTN: phs000517.v3.p1, HT-eMERGE: phs000297.v1.p1, ProC-CGEMS: phs000207.v1.p1, ProC-BPC3: phs000812.v1.p1 and PanC-PanScan: phs000206.v5.p3) used in this study are available at https://www.ncbi.nlm.nih.gov/gap. Data access is governed by the dbGaP Authorized Access program. The genome-wide association datasets (SZ-GAIN, HT-WTCCC: EGAD00000000006, T2D-WTCCC: EGAD00000000009, and healthy control sets: EGAD00000000001, EGAD00000000002) used in this study are available at https://www.wtccc.org.uk/. Data access is controlled by the Wellcome Trust Case Control Consortium. The genome-wide association dataset (SZ-CATIE) is available at https://www.nimhgenetics.org/. Data access is controlled by the NIMH Repository and Genomics Resource (NRGR) support team. All other relevant data is available upon request.

## Code availability

The code for the BridGE (Bridging Gene Sets with Epistasis) method described in this study is available at http://csbio.cs.umn.edu/bridge. This software is freely available for academic use and non-profit research and can be licensed for commercial use.

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

## Acknowledgements

We thank Dr. Frank Albert and Dr. Jing Hou for constructive comments on the manuscript. This work was partially supported by NSF grants DBI 0953881 (C.L.M.) and IIS 0916439 (V.K.), NIH grants R01HG005084 (C.L.M.) and R01HG005853 (C.L.M., C.B.), R01MH097276 (G.F., E.E.S.) and R01GM114472 (G.F.), a University of Minnesota Rochester Biomedical Informatics and Computational Biology Program Traineeship Award (G.F.) and a Walter Barnes Lang Fellowship (G.F.). C.L.M. and C.B. are supported by the CIFAR Genetic Networks program. The content is solely the responsibility of the authors and does not necessarily represent the official views of the funders. Computing resources and data storage services were partially provided by the Minnesota Supercomputing Institute and the UMN Office of Information Technology, respectively. Additional acknowledgements of data sources are included in Supplementary Information.

## Author contributions

G.F., W.W., V.K. and C.L.M. conceived of and designed the method. G.F., W.W., V.P. and B.O. implemented the approach. G.F., W.W., H.H. Xiaoye L., Xiaotong L., B.O. and B.V. performed data processing or method application and evaluation. G.F., W.W., M.C., M.S., B.V.N., E.E.S., N.D.P., C.B., V.K. and C.L.M. analyzed the results of the approach. G.F., W.W., M.C., C.B. and C.L.M. drafted and revised the manuscript with feedback from other co-authors.

## Additional information

**Competing interests:** C.M., M.C. and C.B. are cofounders of, and hold equity in, Bridge Genomics, which is negotiating a licensing agreement with the University of Minnesota for the method used in this paper. The method is described in a pending patent (PCT/US17/22308). The University of Minnesota also has equity and royalty interests in Bridge Genomics. These interests have been reviewed and managed by the University of Minnesota in accordance with its Conflict of Interest policies. The remaining authors declare no competing interests.

