## [Peer Review File · Nature Communications]

Editorial Note: This manuscript has been previously reviewed at another journal that is not operating a transparent peer review scheme. This document only contains reviewer comments and rebuttal letters for versions considered at Nature Communications .

Reviewers' comments:

Reviewer #1 (Remarks to the Author):

Authors have addressed all of my comments satisfactorily.

Reviewer #2 (Remarks to the Author):

The flow and focus are clearer in this version of the manuscript. It is clear you are presenting the BridGE method to globally evaluate pathway-pathway interactions over all of the SNPs in each pathway. That said, it might be better to only include findings that include replication analyses and analyses that address power and Type I error. My specific concerns are outlined below.

1) One thing that should be clarified is how the model is posed. That is, if you say Recessive-Recessive, do you mean that all of the pairs for the SNPs in the 2 pathways should follow this model. Given that the diseases you explore are all complex and very multigenic, it is not clear that such a model is sensible. Perhaps you clarified and possibly justified this in the manuscript and I missed it. There was a lot to follow. If it's in the manuscript, it needs to be moved to or put in a relevant place in the Introduction because it is essential to our understanding of this work.

2)The relevance of BridGE to complex disorders would also be enhanced if the software were made available to investigators. Otherwise, the focus should be on a biological interpretation of your interaction results.

3) As expected, power to detect interactions is improved using BridGE, however Type I errors are not addressed formally.

4) The power analysis is an excellent contribution to the manuscript, and perhaps should be showcased, given that the focus of the manuscript is the BridGE method.

5) Replication of a significant fraction of the Parkinson BPMs is important. It would be helpful to know the probability of the replication you observed, given the large amount of multiple testing and the low SNP-SNP overlap coefficient of .15 Is this reflected by the p of .0001 on line 305. If so, that should be clarified.

6)Given you are presenting the BridGE method as the primary focus here, it might be better to present a tighter argument that it works rather than the extensive amount of pathway interaction information presented for diseases where findings have not been replicated.

Minor points

- 1) There is a difference between 'unlinked' and LD that should be made correct on lines 128 and 129
- 2) It is not clear that 'binarization' is word. Please fix accordingly.

Response to Reviewer 2's comments:

Our response to each comment is included inline below. Changes to the manuscript during this round of revision are highlighted in yellow in the manuscript text.

The flow and focus are clearer in this version of the manuscript. It is clear you are presenting the BridGE method to globally evaluate pathway-pathway interactions over all of the SNPs in each pathway. That said, it might be better to only include findings that include replication analyses and analyses that address power and Type I error. My specific concerns are outlined below.

We appreciate the reviewer's positive feedback on our revised manuscript. Regarding the suggestion about only including replicated findings, we agree that such a change would make our discussion more convincing. In our revised manuscript, we have moved the discussion of discoveries about hypertension and type 2 diabetes (lines 346-357 in previous manuscript), the two diseases for which we did not complete replication analyses, to the supplement (lines 991-1000). We note that our supplemental files still include all discoveries produced by our algorithm on each cohort analyzed, but the discoveries that replicate are clearly marked in these files.

1) One thing that should be clarified is how the model is posed. That is, if you say Recessive-Recessive, do you mean that all of the pairs for the SNPs in the 2 pathways should follow this model. Given that the diseases you explore are all complex and very multigenic, it is not clear that such a model is sensible. Perhaps you clarified and possibly justified this in the manuscript and I missed it. There was a lot to follow. If it's in the manuscript, it needs to be moved to or put in a relevant place in the Introduction because it is essential to our understanding of this work.

We apologize for the lack of clarity in how we described the disease model in the main text. To clarify, when run in the Recessive-Recessive mode, our approach does test all SNP pairs across the two pathways using the same disease model. However, our method also has a "combined" mode, which integrates edges derived from multiple disease models, which relates to the reviewer's concern. The combined model allows for SNPs to interact under multiple disease models as long as they still connect across the two pathways of interest.

To make this more clear, we added a sentence in the main text (lines 133-136) near our discussion of the disease model to clarify these different modes. We also have additional details regarding the choice of disease model in the supplementary information (see "SNP-SNP genetic interaction estimation" section), which is also cited in the relevant section of the main text.

"BridGE can use any one of four different disease models to compute a SNP-SNP interaction network: additive, recessive, dominant, or combined recessive and dominant. When run in the additive, recessive and dominant mode, BridGE tests all SNP pairs across the two pathways using the same disease model. In the combined recessive and dominant mode, BridGE integrates edges derived from multiple disease models, which allows for SNPs to interact in multiple ways as long as they still connect across the two pathways of interest. Each model encodes different assumptions and will result in a different, overall complementary, set of significant BPM/WPM structures at the end of the pipeline."

We note that the recessive-recessive model produces fewer results than either the combined or dominant model (the Recessive-recessive model produces results in 1 out of 13 cohorts), which is partially due to the issue anticipated by the reviewer, but also simply a consequence of reduced

power due to lower allele frequency. We have discussed this observation in more detail in Supplementary Method section “12. Dependence of interaction discoveries on the assumed disease model”.

2)The relevance of BridGE to complex disorders would also be enhanced if the software were made available to investigators. Otherwise, the focus should be on a biological interpretation of your interaction results.

Yes, we agree. The software is publicly available and was linked in our original submission, <http://csbio.cs.umn.edu/bridge> (Line 175).

3) As expected, power to detect interactions is improved using BridGE, however Type I errors are not addressed formally.

We are confused by the reviewer’s comment about Type I error as we addressed this specific point in our last round of revisions. Our analysis throughout the manuscript reports the False Discovery Rate (FDR) for all discoveries we describe, which controls Type I errors in multiple testing settings. If this comment is regarding how Type I errors are addressed in the power simulations, we also addressed this during the last round of comments (see below):

Response copied from our previous submission:

Our power analysis does account for both Type I and Type II error. To simplify the presentation of these results, for each scenario we explored (e.g. pathway size, interaction effect size, minor allele frequency), we identified the minimum cohort size that would be sufficient to detect BPMs with the corresponding characteristics at a fixed false discovery rate (which controls the Type I error). So these simulations directly reflect the sensitivity (sensitivity = $[1 - \text{Type II error}]$) in discovering BPMs under the practical scenario where Type I error rate is controlled (through control of the FDR). We have expanded our description of the power analysis to clarify that it addresses both Type I and Type II error.

We note that our power simulation is necessarily different than the canonical approach used to measure power for single-variant detection methods or even for SNP-level analyses of interactions. The fact that our method discovers pathway-level interactions demands a unique approach to power simulation.

Sentence added to the main text:

“For each parameter setting, we measured the minimum cohort size required to detect a pathway-level interaction with the corresponding properties at a fixed FDR ($\text{FDR} < 0.25$) (i.e. controlling the Type I error rate).”

Sentence added to the Methods section (Section 13):

“We note that our power analysis accounts for both Type I and Type II error. Specifically, the simulations directly reflect the sensitivity (sensitivity = $[1 - \text{Type II error}]$) as a function of

sample size in discovering BPMs under the practical scenario where Type I error rate is controlled (through control of the FDR) for exhaustive pairwise tests for BPMs.”

4) The power analysis is an excellent contribution to the manuscript, and perhaps should be showcased, given that the focus of the manuscript is the BridGE method.

We thank the review for the positive feedback about our power analysis. As suggested, we have now expanded the description of our power analysis in the main text (new expanded paragraph now appears from lines 363-375 in the new version). Additional details on this process are still described in the Online Methods to complement the description in the manuscript main text as the process is quite technical.

5) Replication of a significant fraction of the Parkinson BPMs is important. It would be helpful to know the probability of the replication you observed, given the large amount of multiple testing and the low SNP-SNP overlap coefficient of .15 Is this reflected by the p of .0001 on line 305. If so, that should be clarified.

We thank the reviewer for this comment. We agree that controlling for multiple testing during replication analysis is critical, and our replication analysis was designed to specifically account for the number of discoveries we attempted to replicate across studies.

Specifically, we tested all BPMs/WPMs derived from each discovery dataset for significance in the replication cohort and then compared the number of those meeting a given significance threshold to the average number of replications observed across 100 random sample permutations of the replication dataset. This approach controls for the number of replications attempted and provides a significance measure for the observed level of overall replication. For example, for Parkinson’s disease, the resulting p-value from this replication test was $p=0.02$ (reported on line 266, Supplementary Table S6); for Prostate Cancer this was $p=0.0001$ (lines 308 and 310), for Breast cancer, $p=0.07$ (lines 311); for Schizophrenia, $p=0.02$ and $p=0.03$ (lines 314-315) (breast cancer, prostate cancer, and schizophrenia results included in Supplementary Table S19).

We should note that there is a slight misunderstanding regarding the SNP-SNP overlap coefficient analysis. This analysis is distinct from the replication analysis described above. A BPM or WPM could replicate in an independent study but could possibly be supported by different individual SNP-SNP pairs. To explore if this was the case, we explored the overlap between SNP-SNP interactions between the two Parkinson’s disease datasets for BPMs that replicated, and observed that this overlap was relatively modest (0.15 as mentioned by the reviewer), but statistically significant. This result suggests that while some of the individual SNP-SNP interactions are in common across the two populations, there is likely distinct signal at the individual SNP level even where pathway-level genetic interactions are shared.

6) Given you are presenting the BridGE method as the primary focus here, it might be better to present a tighter argument that it works rather than the extensive amount of pathway interaction information presented for diseases where findings have not been replicated.

A previous version of our manuscript was focused only on our Parkinson's disease results and placed more emphasis on the method itself rather than its application to many different diseases. At that time, reviewers and the journal editor coordinating the review requested that in order to convince readers that our approach indeed generalizes to many human traits, we should expand our application to many other diseases and describe the results. This is the reason why our current manuscript now presents results on 7 different diseases, including discoveries on 6 of these 7 diseases. We understand the reviewer's perspective, but still feel that presenting evidence that our approach works in many different disease settings is important for demonstrating the generalizability of the method. Furthermore, discussing the biological relevance of a subset of our discoveries relative to what is known about disease mechanisms is important for demonstrating the novel and relevant insights our approach can produce.

Minor points

1) There is a difference between 'unlinked' and LD that should be made correct on lines 128 and 129

We modified the “a subset of unlinked SNPs “ to “a subset of SNPs that are less likely in LD”.

2) It is not clear that 'binarization' is word. Please fix accordingly.

The word 'binarization' is mostly used in network-based approaches and has been widely accepted by scientific journals, including Nature Communications (e.g. Huynh et al. 2016, Pietsch et al. 2016, Watanabe et al. 2017).

REVIEWERS' COMMENTS:

Reviewer #2 (Remarks to the Author):

All of my concerns have been clearly and carefully addressed.